



# Exploring the Potential of History Matching for Land Surface Model Calibration

Nina Raoult[1], Simon Beylat[2,3], James M. Salter[1], Frédéric Hourdin[4], Vladislav Bastrikov[5], Catherine Ottlé[2], and Philippe Peylin[2]

[1]Department of Mathematics and Statistics, Faculty of Environment, Science and Economy, University of Exeter, Laver Building, North Park Road, Exeter, EX4 4QE, UK
[2]Laboratoire des Sciences du Climat et de l'Environnement, LSCE/IPSL, CEA-CNRS-UVSQ, Université Paris-Saclay, Gif-sur-Yvette, 91191, France
[3]School of Geography, Earth and Atmospheric Sciences, University of Melbourne, Victoria, Australia
[4]Laboratoire de Météorologie Dynamique, LMD/IPSL, Sorbonne Université, CNRS, École Polytechnique, ENS, Paris, 75005, France
[5]Science Partners, Paris, France

**Correspondence:** Nina Raoult (n.m.raoult2@exeter.ac.uk)

**Abstract.**

With the growing complexity of land surface models used to represent the terrestrial part of wider Earth system models, the need for sophisticated and robust parameter optimisation techniques is paramount. Quantifying parameter uncertainty is essential for both model development and more accurate projections. In this study, we assess the power of history matching by

comparing results to variational data assimilation, commonly used in land surface models for parameter estimation. Although both approaches have different setups and goals, we can extract posterior parameter distributions from both methods and test the model-data fit of ensembles sampled from these distributions. Using a twin experiment, we test whether we can recover known parameter values. Through variational data assimilation, we closely match the observations. However, the known parameter values are not always contained in the posterior parameter distribution, highlighting the equifinality of the parameter space.

In contrast, while more conservative, history matching still gives a reasonably good fit and provides more information about the model structure by allowing for non-Gaussian parameter distributions. Furthermore, the true parameters are contained in the posterior distributions. We then consider history matching's ability to ingest different metrics targeting different physical parts of the model, helping to reduce parameter space further and improve model-data fit. We find the best results when history matching is used with multiple metrics; not only is the model-data fit improved, but we also gain a deeper understanding of the

model and how the different parameters constrain different parts of the seasonal cycle. We conclude by discussing the potential of history matching in future studies.

## 1 Introduction

Land surface models (LSMs) are essential for studying land-atmosphere interactions and quantifying their impact on the global climate. They help us comprehend and represent the mass and energy fluxes exchanged in the soil-vegetation-atmosphere





continuum, as well as the lateral transfers. However, despite their increasing complexity (Fisher and Koven, 2020), these
models remain subject to large uncertainties, in terms of missing processes and poorly constrained parameters. Reducing this
uncertainty is crucial to generate reliable and credible model projections, especially since creating robust predictions of the
terrestrial biosphere is becoming a critical scientific priority, e.g. in the context of plans for land-based climate mitigation such
as re-greening (Roe et al., 2021).

To address this parametric uncertainty, it is customary to calibrate (or "tune") the model. This means finding model parame-
ters that provide a good description of the system's behaviour, often taken to be the model's ability to reproduce observations.
In LSMs, this is commonly achieved through data assimilation (DA), which uses a Bayesian framework to account for prior
parameter knowledge and to obtain posterior values and uncertainties. DA can be used to improve the initial state of the model
and/or the internal model parameters. In numerical weather forecasting, where the fundamental equations are mostly known

(e.g., Navier-Stokes), DA is predominately used to correct the initial state. In contrast, in climate studies, we rely less on ini-
tial state optimisation and more on parameter calibration, especially for the carbon cycle, where a lot of processes are based
on empirical equations. Furthermore, we often rely on variational DA (VarDA) approaches, in which all observations within
the assimilation time-window are used to create a cost function which is then minimised. Over the last 15 years, VarDA has
been successfully used in land surface modelling to optimise uncertain parameters (MacBean et al., 2022). The focus of these

optimisations has often been to better estimate carbon stocks and fluxes (Kuppel et al., 2012; Kaminski et al., 2013; Raoult
et al., 2016) by targeting vegetation and carbon cycle related parameters, although more some recent studies have also focus
on improving LSM soil moisture predictions (Scholze et al., 2016; Pinnington et al., 2018; Raoult et al., 2021). However, most
of these examples use a limited number of in situ data to calibrate a handful of parameters. As LSMs become more complex,
so must the experiments used to calibrate them. Increased process representation and the tighter coupling between the different

terrestrial cycles (e.g., water, carbon and energy) mean that more parameters need to be considered. As a result, LSMs are
also becoming more costly to run. Furthermore, although satellite retrievals now provide us with data in previously hard to
monitor areas, these data at high temporal and spatial resolutions need to be carefully ingested and contribute to more costly
optimisations.

Emulators, i.e., simplified or surrogate models that are used to approximate complex model behaviour, can provide a solution

to some of these computational challenges. They are constructed by interpolating between the points where the model has been
run. Indeed, emulator-based LSM parameterisation has been gaining traction in recent years (Fer et al., 2018; Dagon et al.,
2020). Emulators can be used to emulate LSM outputs (e.g., Kennedy et al., 2008; Petropoulos et al., 2014; Huang et al., 2016;
Lu and Ricciuto, 2019; Baker et al., 2022). However, this can be challenging given the large, nonlinear multivariate output
space. Fortunately, for calibration, we do not need to emulate the full output space, but rather the property we seek to improve

- for example, the likelihood (Fer et al., 2018), which has the advantage of generally being smooth and univariate (Oakley and
Youngman, 2017).

The rise of emulators in the field of LSM calibration has also led to the preliminary testing of the so-called history matching
(HM) method to tune LSM parameters (Baker et al., 2022; McNeall et al., 2023). This is a different approach which asks
not what is the best set to use but, rather, what parameters can we rule out: what regions of parameter space lead to model



outputs being "too far" from observations? To do this, HM uses an implausibility function, based on metrics chosen to assess the performance of the model, to rule-out unlikely parameters. HM commonly uses an iterative approach known as iterative refocusing to reduce parameter space, leaving the least unlikely parameter values - the not-ruled out yet (NROY) space. This is a more conservative approach to calibration, primarily used for uncertainty quantification, helping to identify structural deficiencies of the model (Williamson et al., 2015; Volodina and Challenor, 2021). Although this technique can work without

emulators (i.e., if the model is extremely fast, Gladstone et al. (2012)), the high cost of running LSMs means that emulators will likely be required.

HM has successfully been used in a number of fields, making it an established statistical method with a diverse literature. Initially, it was introduced as a method for discovering parameter configurations for computationally intensive oil well models (Craig et al., 1997). It has since been used in various domains of science and engineering, such as galaxy formation (Bower

et al., 2010; Vernon et al., 2014), disease modelling (Andrianakis et al., 2015), systems biology models (Vernon et al., 2022), and traffic (Boukouvalas et al., 2014). In climate sciences, HM was also used to calibrate climate models of different complexities (Edwards et al., 2011; Williamson et al., 2013, 2015; Hourdin et al., 2023), ocean models (Williamson et al., 2017; Lguensat et al., 2023), atmospheric models (Couvreux et al., 2021; Hourdin et al., 2021; Villefranque et al., 2021) and ice sheet models (McNeall et al., 2013).

Here, we present the application of HM to an LSM, starting with its implementation into the ORCHIDEE Data Assimilation System (ORCHIDAS; https://orchidas.lsce.ipsl.fr/). Using a twin experiment with known model parameters and model errors, we explore HM's ability to recover these parameters and the resulting model fit to the data (here the model run with the true parameters). There are two parts to this study. In the first part, we compare HM to VarDA by considering the two minimisation techniques historically used to calibrate the ORCHIDEE LSM (i.e., a gradient-based and a Monte Carlo approach). We initially

use a root-mean-square difference metric in the HM experiment to mimic the cost function used in VarDA. Given the different motivations behind VarDA and HM, we are less interested in finding the optimal set of parameters but rather in whether the true parameters are contained in the posterior distributions obtained and how the spread of the model runs generated from sampling those distributions fits the data. In the second part of the study, we delve deeper into the HM methodology to demonstrate its versatility in considering different target metrics. We test whether we can more closely constrain parameters involved in

different processes by specifically targeting these processes with our metrics. We conclude by discussing our study's limitations and exploring future avenues for employing HM in LSM calibration.

## 2  Methods and Data

### 2.1  ORCHIDEE land surface model

The ORCHIDEE (ORganizing Carbon and Hydrology In Dynamic EcosystEms; originally described in Krinner et al. (2005))

model simulates the carbon, water and energy exchanges between the land surface and the atmosphere. Fast processes such as photosynthesis, hydrology and energy balance are computed at a half-hour time step, while slow processes such as carbon allocation and phenology are simulated daily. The model can be run at different resolutions ranging from point scale to global,



offline (i.e., with meteorological forcing data externally applied) or coupled as part of the wider IPSL (Institut Pierre Simon Laplace) Earth system model. In this study, we use version 2.2 of the ORCHIDEE model, which is the one used in the Coupled

Model Intercomparison Project Phase 6 (CMIP6; Boucher et al., 2020; Lurton et al., 2020).

## 2.2 Data Assimilation Framework - the ORCHIDAS system

The ORCHIDAS system is set up to optimise the parameters of the ORCHIDEE model. It has been used in over 15 years of terrestrial optimisation studies (MacBean et al., 2022), initially with a focus on the carbon cycle and more recently used to optimise parts of the other terrestrial cycles such as water (Raoult et al., 2021), methane (Salmon et al., 2022) and nitrogen

(Raoult et al., 2023). See orchidas.py for a full list of published studies.

This flexible framework easily allows ORCHIDEE to be run with many different parameter settings, which historically are used to minimise a cost function (using a standard Bayesian calibration setup) or test the sensitivity of the model using classic sensitivity analysis methods (e.g., Morris and Sobol). For this study, a HM methodology adapted from Hightune (the LMDZ HM tool developed to improve and calibrate the parameterisations involved in the representation of boundary layer clouds,

Couvreux et al. (2021); Hourdin et al. (2021); Villefranque et al. (2021)) was added to ORCHIDAS, allowing these different runs to be used to train emulators and used to calculate implausibility.

### 2.2.1 A Bayesian setup

We use a Bayesian setup to account for model and observation errors. Therefore, we need to establish how we statistically model the relationship between the observations and the model variables. Following Kennedy and O'Hagan (2001)'s best input

approach, for an observational constraint $\mathbf{z}$, let

$$\mathbf{z} = \mathbf{y} + \mathbf{e}, \tag{1}$$

where $\mathbf{y}$ represents the underlying aspects of the system being observed, and $\mathbf{e}$ represents uncorrelated error on these observations (perhaps comprising instrument error and any error in deriving the data products making up $\mathbf{z}$). Note that this observation error, $\mathbf{e}$, is treated as a random quantity with mean 0 and variance $\sigma_e^2$ (i.e., $\mathbf{e} \sim \mathcal{N}(0, \sigma_e^2)$). We then assume that $\mathbf{x}^*$ is the 'best

input' to our model $H$ and with $\boldsymbol{\eta}$ denoting the model discrepancy, we get:

$$\mathbf{z} = \mathbf{y} + \mathbf{e} = H(\mathbf{x}^*) + \boldsymbol{\eta} + \mathbf{e}. \tag{2}$$

The model discrepancy, which is assumed to be independent of $\mathbf{x}^*$ and $H(\mathbf{x})$, accounts for the model structural error due to the inherent inability of the model to reproduce the observations exactly (e.g., due to unresolved physics or missing processes, parameterisation schemes, resolution of numerical solvers). This error has mean 0 (unless the user knows the direction in which

the model is biased) and variance $\sigma_\eta^2$ (i.e., $\boldsymbol{\eta} \sim \mathcal{N}(0, \sigma_\eta^2)$).





### 2.2.2 Variational data assimilation

In variational data assimilation (VarDA), we are looking for $p(\mathbf{x}|\mathbf{z})$, i.e., the distribution of parameters given the observations. Given a known parameter vector called the background (or prior, $\mathbf{x}_b$), the knowledge of parameters is described by the probability density function $p(\mathbf{x})$. Similarly, $p(\mathbf{z}|\mathbf{x})$ is the likelihood of the observations $\mathbf{z}$ given the the parameters $\mathbf{x}$. Bayes' theorem can be used to combine these probabilities

$$p(\mathbf{x}|\mathbf{z}) \propto p(\mathbf{z}|\mathbf{x})p(\mathbf{x}). \tag{3}$$

Gaussian distributions are commonly used to represent the different terms of the optimisation, so that:

$$p(\mathbf{z}|\mathbf{x}) \propto \exp\left[-\frac{1}{2}(\mathbf{z} - H(\mathbf{x}))^T\mathbf{R}^{-1}(\mathbf{z} - H(\mathbf{x}))\right]; \qquad p(\mathbf{x}) \propto \exp\left[-\frac{1}{2}(\mathbf{x} - \mathbf{x_b})^T\mathbf{B}^{-1}(\mathbf{x} - \mathbf{x_b})\right], \tag{4}$$

where $\mathbf{R}$ and $\mathbf{B}$ are the covariance matrices of the observation/model errors (i.e., $e + \eta$) and the background errors, respectively.

When combining these analytical expressions, we find that maximising the likelihood of $p(\mathbf{x}|\mathbf{z})$ is equivalent to minimising the cost function:

$$J(\mathbf{x}) = \frac{1}{2}\left[(H(\mathbf{x}) - \mathbf{z})^T\mathbf{R}^{-1}(H(\mathbf{x}) - \mathbf{z}) + (\mathbf{x} - \mathbf{x}_b)^T\mathbf{B}^{-1}(\mathbf{x} - \mathbf{x}_b)\right]. \tag{5}$$

Note that this is known as finding the maximum a posterior probability estimate in Bayesian statistics. Many algorithms can be used to minimise this cost function. They broadly fall into two methods: deterministic gradient-based methods and stochastic random search methods. Here we consider one from each category, both of which are commonly used in land surface model parameter estimation studies. The first is the quasi-Newton algorithm L-BFGS-B (limited memory Broyden–Fletcher–Goldfarb–Shanno algorithm with bound constraints; see Byrd et al. (1995)), henceforth referred to as BFGS. The second is the genetic algorithm (GA; Goldberg and Holland (1988); Haupt and Haupt (2004)) based on the laws of natural selection and belongs to the class of evolutionary algorithms. It considers the set of parameters as a chromosome, with each parameter as a gene. At each iteration, the algorithm generates a population $g$ of chromosomes by recombining and possibly randomly mutating (defined by a mutation rate) the fittest chromosomes from the previous iterations. Both methods are fully described in Bastrikov et al. (2018).

With the assumed Gaussian prior errors and further assuming linearity of the model in the vicinity of the solution we can approximate the posterior covariance error matrix $\mathbf{B}_{\text{post}}$:

$$\mathbf{B}_{\text{post}} = \left[\mathbf{H}^T\mathbf{R}^{-1}\mathbf{H} + \mathbf{B}^{-1}\right]^{-1} \tag{6}$$

where $\mathbf{H}$ is the model sensitivity (Jacobian) at the minimum of the cost function (Eq. 5; see Tarantola (2005)). To estimate the posterior uncertainty of the parameters, we sample from the multivariate normal distribution $\mathcal{N}(\mathbf{x}_{\text{opt}}, \mathbf{B}_{\text{post}})$ (Tarantola, 2005, Chapter 3.3.1). This ensures that the whole $\mathbf{B}_{\text{post}}$ matrix is used, including off-diagonal elements that describe the covariance between parameters. Since $\mathbf{B}_{\text{post}}$ relies on information about the curvature of parameter space, it lends itself well to gradient-based approaches (e.g. BFGS). Nevertheless, it can be used to calculate posterior distribution at the end of any optimisation



algorithm. GAs, although a Monte Carlo technique, lack the basic theorem of the Metropolis algorithm (Tarantola, 2005), which involves sampling the parameter space according to a prescribed distribution. Instead, genetic algorithms follow unknown distributions and, therefore, cannot be used directly for the Monte Carlo integration (Sambridge, 1999) needed to calculate the posterior parameter distribution. While methods do exist to resample parameter space, these are not without limitations and are out of the scope of this study. Instead, we also calculate $\mathbf{B}_{\text{post}}$ at the end of the GA optimisations, using the same hypothesis as for BFGS above.

In this work, we use a diagonal $\mathbf{R}$ matrix, and therefore we can think of the matrix $\mathbf{R}$ as a vector of errors $\boldsymbol{\sigma}$. This means that when performing a multi data-stream optimisation (i.e. when considering more than one variable), we can decompose the first term of $\mathbf{B}_{\text{post}}$ in the following manner

$$\mathbf{H}^T \mathbf{R}^{-1} \mathbf{H} = \sum_{i=0}^{D} \mathbf{H}^T \sigma_i^{-1} \mathbf{H} \tag{7}$$

where $D$ is the total number of data streams used in the optimisation and $\sigma_i$ is the error associated to each data stream. Using this decomposition, we can create some proxy posterior covariance matrices associated with each flux:

$$\mathbf{B}'_{\text{post}_i} = [\mathbf{H}^T \sigma_i^{-1} \mathbf{H} + \mathbf{B}^{-1}]^{-1}, \tag{8}$$

to get an insight into the different constraints each separate set of observations has on the posterior parameters.

### 2.2.3 History matching

In Bayesian history matching (HM), we use observed data to rule out any parameter settings that are "implausible", usually done with the help of an emulator. It commonly uses the notion of iterative refocusing, where model simulations at each iteration (referred to as a wave) are chosen to improve the emulator and the calibration. Instead of using a unique cost function, an implausibility is computed independently for different metrics to rule out parameters too far from the target:

$$\mathcal{I}(\mathbf{x}) = \frac{|\mathbf{z} - \mathrm{E}[H(\mathbf{x})]|}{\sqrt{\mathrm{Var}[\mathbf{z} - E[H(\mathbf{x})]]}} \tag{9}$$

$$= \frac{|\mathbf{z} - \mathrm{E}[H(\mathbf{x})]|}{\sqrt{\mathrm{Var}[H(\mathbf{x})] + \mathrm{Var}[\mathbf{e}] + \mathrm{Var}[\boldsymbol{\eta}]}}. \tag{10}$$

Large values of $\mathcal{I}(\mathbf{x_i})$ for a given $\mathbf{x_i}$ implies that, relative to our uncertainty specification, it is implausible that $H(\mathbf{x_i})$ is consistent with the observations and therefore $\mathbf{x_i}$ can be ruled out. Note that to calculate the implausibility, we only require the observation error variance, the model discrepancy variance, and the variance and expectation of $H(\mathbf{x})$, which can be defined using an emulator. Unlike in VarDA, there is no background term, meaning we do not need to calculate a $\mathbf{B}$ matrix similar to the one found in Eq. 5.

By choosing a threshold $a$, we can formally define the not ruled-out yet (NROY) space as:

$$\mathcal{X}_{\text{NROY}} = \{\mathbf{x} \in \mathcal{X} : \mathcal{I}_m(\mathbf{x}) \leq a, \forall m\}. \tag{11}$$





where $\mathcal{X}_m$ is the implausibility for a given metric $m$. The value of $a$ is often taken to be 3 following the $3\sigma$ rule (Pukelsheim, 1994). This states that for any unimodal continuous probability distribution, at least 95% of the probability mass is within three standard deviations of the mean.

To increase computational efficiency, it is very common to use emulators in HM. Here, we use Gaussian Process (GP) emulators - a well-known statistical model that has the advantage of interpolating observed model runs and provides a probabilistic prediction (and hence variance) for the model at unseen $\mathbf{x}$, which is required for the implausibility computation (Couvreux et al., 2021). The emulator gives the following probability distribution for $H$:

$$H(\mathbf{x})|\boldsymbol{\beta}, \sigma^2, \boldsymbol{\delta} \sim \mathrm{GP}\left(m(\mathbf{x};\boldsymbol{\beta}), k(\cdot, \cdot, \sigma^2, \boldsymbol{\delta})\right) \tag{12}$$

where $m(\mathbf{x};\boldsymbol{\beta})$ is a prior mean function with parameters $\boldsymbol{\beta}$ and $k$ is a specified kernel (i.e., a covariance function). Within the kernel, the variance is controlled by $\sigma^2$, and each element of $\boldsymbol{\delta}$ controls the correlation attributed to each input. These emulators are trained on the true model runs. For more specifics about how the emulators are built, see Williamson et al. (2013).

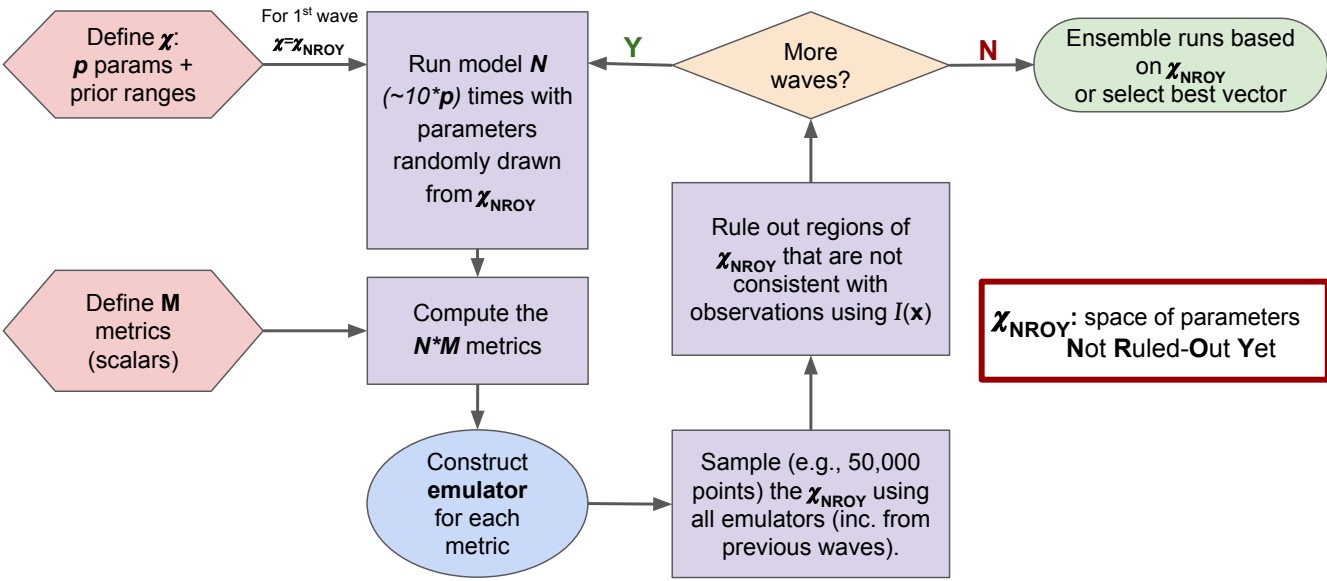

**Figure 1.** A flowchart showing the HM process in ORCHIDAS.

Figure 1 illustrates how HM is used in the ORCHIDAS system. We first define our $p$-dimensional parameter space $\mathcal{X}$. From this space, parameter sets are randomly drawn, at which the model is run. It is common to run the model approximately 10 times the number of parameters (Loeppky et al., 2009). The outputs from each run are then mapped onto scalar values with a metric (e.g., root-mean-square deviation, RMSD). Emulators are constructed for each metric and these are used to sample from $\mathcal{X}_{\mathrm{NROY}}$. This allows us to have a lot more points than model runs in the ruling-out step. The implausibility (Eq. 10) is used to rule out points, refining the $\mathcal{X}_{\mathrm{NROY}}$. More waves can then be conducted using this new space until we are satisfied with the remaining space. If the $\mathcal{X}_{\mathrm{NROY}}$ is empty after a wave, it means that we cannot match the observations given the current error





tolerances. If the $\mathcal{X}_{\text{NROY}}$ no longer reduces between successive waves, then it might signify that the emulator variance is too high relative to the spread of the ensemble. As such, training the emulator with more runs may be necessary. It may also mean that the system has converged and all remaining points are within the tolerance set. As a result, we may be able to reduce the

cutoff ($a$). Although in this study we only focus on the same metrics throughout each HM experiment, there is the potential to change them as the waves progress.

## 2.3 Experimental set up

### 2.3.1 Twin experiment

We perform a twin experiment to compare the different approaches in a controlled manner. This means we generate a set of

pseudo-observations using a 'true' set of parameters; here, we use the default ORCHIDEE values. Gaussian white noise, with a standard deviation set to 0.1 times the time series' mean, was added to each timestep to represent the model/observation error. We use this error to set up the experiments: $\mathbf{e} + \boldsymbol{\eta}$ (with $\boldsymbol{\eta} = 0$) for HM and the diagonal element for the $\mathbf{R}$ in VarDA. We further use a diagonal $\mathbf{B}$ in our VarDA experiments, where prior uncertainty is set to 100% of the parameter range of variation to allow for maximal space exploration.

We focus our study on a temperate broadleaf deciduous forest site from the eddy-covariance Fluxnet database (Pastorello et al., 2020), FR-Fon (Fontainebleau-Barbeau; Delpierre et al. (2016)). This site is often used in our calibrations. Here, we focus on the first year of the time series (year 2005) for calibration and the rest of the time series (years 2006-2009) for evaluation. Since we are running a twin experiment and, therefore, the observations are artificially generated, we only use the Fluxnet meteorological data to drive our model. As in previous work (e.g., Kuppel et al., 2012; Bastrikov et al., 2018),

we focus on the model's ability to simulate net ecosystem exchange (NEE) and latent heat (LE) fluxes. NEE represents the difference between carbon dioxide uptake by plants through photosynthesis and carbon release through respiration, with the growing season typically characterised by negative NEE indicating net carbon absorption. LE represents the exchange of energy between the Earth's surface and the atmosphere through the phase changes of water, with higher values during periods of increased evaporation and transpiration, often associated with warmer seasons. The parameters for this study were chosen

with these fluxes in mind, using our past expertise, a preliminarily Morris (Morris, 1991) sensitivity analysis, and the desire to work with a small set of parameters (see Supplementary Material for more on the preliminary sensitivity tests). These parameters are listed in Table 1.

### 2.3.2 Performed experiments

To minimise the cost function (Eq. 5) in the VarDA experiments, we consider both BFGS (local gradient descent) and GA

(global random search) optimisation techniques. For BFGS, the algorithm is run for 25 iterations, which was found sufficient for the optimisation to converge. For GA, a population of 24 and a mutation rate of 0.2 were used along with 25 iterations. These values are based on previous optimisations performed using Fluxnet data to optimise simulated NEE/LE in the ORCHIDEE model (Bastrikov et al., 2018) and were also found to be sufficient for convergence.





**Table 1.** ORCHIDEE parameters used in this study. True value refers to the default value of each parameter in ORCHIDEE. These values were used to generate the observations used in the twin experiment. Range refers to the range of variation allowed for each parameter.

| | Description | True value | Range |
|---|---|---|---|
| $VC_{max}$ | Maximum carboxylation rate ($\mu molm^{-1}s^{-1}$) | 50 | [30, 80] |
| SLA | Specific leaf area ($m^2$) | 0.026 | [0.013, 0.05] |
| $L_{agecrit}$ | Critical leaf age for starting leaf senescence (days) | 180 | [90, 240] |
| $Evap_{res}$ | Factor controlling bare soil resistance to evapotranspiration (-) | 1 | [0, 1.3] |
| $Root_{prof}$ | Root profile parameter of an exponential function that describes the decrease of root density as a function of depth (m) | 0.8 | [0.2, 3.0] |
| $Q_{10}$ | Parameter determining the temperature dependency of the heterotrophic respiration (-) | 0.69 | [0, 1.1] |

We perform two sets of experiments per minimisation algorithm. The first set of experiments uses $\mathbf{B}_{post}$ to assess posterior

distributions after a single optimisation. To calculate the posterior uncertainties, we sample 1e4 points from $\mathcal{N}(\mathbf{x}_{opt}, \mathbf{B}_{post})$ using the `random.multivariate_normal` function from the NumPy python package (Papoulis, 1991; Duda et al., 2001). In the second set of experiments, we perform many optimisations (200), starting from random priors, and use the posterior parameter values to elicit the posterior parameter uncertainty. Given the probabilistic results obtained, we refer to these experiments as "stochastic". In a standard optimisation, we would use the default model parameter values as prior, since they are

our best guess. However, as we are performing a twin experiment where the default parameter values are the true solution, we must start from a different part of the parameter space. To do this, we generate several random parameter sets. For the $\mathbf{B}_{post}$ experiments, where we consider only one optimisation, we chose the randomly generated parameter set that starts closest to the true values to be the most realistic. Although not shown, we repeated the analysis using a different prior, which gave similar results. For the stochastic experiments, we start from 200 different uniformly and randomly generated priors.

For HM, we do not need to worry about prior parameter values, only the parameter ranges. For each wave, the model is run 60 (i.e., 10 times the number of parameters) times. Initially, we consider the RMSD between the model and true model run as the target metric, since this closely relates to the cost function used in VarDA (Eq. 5). In the second part of the study, we vary the metrics to fully explore the power of HM. We perform five experiments, four using different metrics (listed in Table 2) and the fifth combining all the metrics. We perform ten waves in each case and keep a constant cutoff of 3. At each step, we check the emulator quality (see Sect. B). We also retain the true model runs that are below the cutoff to train the next emulators.





**Table 2.** Target and variance used for each metric tested in the HM experiments. Min/Max taken as the minimum/maximum of the smoothed annual cycle (12-period rolling mean with a window size of five). Spring slope is taken as the difference between April and February monthly means, Autumn slope is taken to be the difference between September and August monthly means, and the initial C stocks are taken as the starting value of the time series.

| Metric | NEE ($gCm^{-1}d^{-1}$) | LE ($Wm^{-2}$) |
|---|---|---|
| RMSD | $0 \pm 0.04$ | $0 \pm 12.5$ |
| Min/Max | $-4.81 \pm 0.004$ | $101.96 \pm 0.696$ |
| Spring slope | $-5.46 \pm 0.003$ | $59.53 \pm 0.72$ |
| Autumn slope | $2.16 \pm 0.004$ | $-29.87 \pm 0.85$ |
| Initial C stocks | $1.558 \pm 0.0002$ | N/A |

# 3   Results

## 3.1   Comparing variational data assimilation and history matching

### 3.1.1   Model-data fit

The first step in any calibration experiment is commonly to check the posterior model-data fit. Instead of considering the
fit given by a single parameter set for each experiment, we consider the ensemble of posterior parameters taken from each experiment (Fig. 2). For the VarDA results, we consider model runs generated from each optimal parameter set found in the stochastic experiments (i.e., 200 optimisations) since these results give a larger posterior spread than the $\mathbf{B}_{\text{post}}$ experiments (not shown). For HM, we consider the experiment using the RMSD as the target metric since this most closely relates to the cost function used in VarDA. For consistency, we consider 200 parameter sets sampled from the $\mathcal{X}_{\text{NROY}}$ found at the end of the
experiment (i.e., after ten waves) and use these to run the model to create the posterior ensembles.

    The model-data fit in all experiments is much improved compared to the prior ensemble - with the GA experiment closely matching the observations, BFGS performing second best, and HM retaining the most spread. Overall, we are able to capture the seasonality and general magnitude of the observed fluxes. Typically, the only parts of the observations that are not contained in the posterior spread are the LE winter values, which are also outside of the prior ensemble spread. This suggests that these
values are not reproducible by the model and represent the structural error (i.e., the noise we added to the true model realisation used to generate these pseudo-observations). We also note that the GA experiment gives an ensemble spread smaller than the spread of the noise on the observations, suggesting that these parameter sets may have overfitted the data.

    Although the general shape of the seasonal pattern is reasonably well matched, we can see there are parts of the time series we are less able to constrain, especially with the HM experiment. The slope in spring, for example (also noticeable for BFGS),
and the behaviour in summer. This highlights the issue of relying on a single metric in the optimisation process to capture the full behaviour of a time series, particularly the RMSD. The RMSD is prone to correct large errors and therefore may be



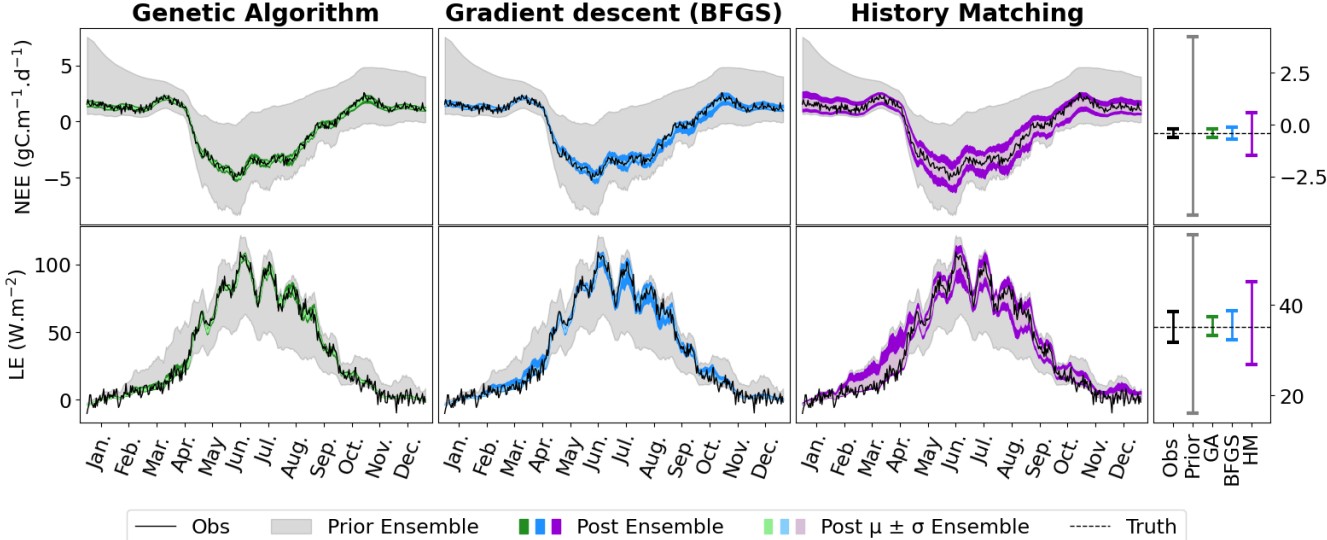

**Figure 2.** Time series of NEE (top) and LE (bottom) for FR-Fon year 2005. For the VarDA experiments (minimisation methods GA and gradient-descent BFGS), the spread represents the results taken from the stochastic experiments (i.e., from 200 optimisations). For the HM experiment, 200 parameter sets were sampled from the $\mathcal{X}_{\mathrm{NROY}}$ and used to run the model. The prior ensemble, i.e., before any calibration, is shown in grey and the posterior ensemble is shown in dark colours. The lighter coloured spread shows the mean and standard derivation of the posterior ensemble. The bar plots on the right-hand side show the range of the mean of each ensemble time series. Note the difference in scales.

strongly driven by outliers. As such, it works well at correcting the errors in amplitude but less well at fitting other temporal features.

Figure 2 shows the fit to the calibration year - i.e., the one used to tune the model. We also tested the ensembles over several

more years (2006-2009) to further evaluate the results. This evaluation step is important to check that we do not over-tune to the specificities of a given year but find parameter sets that work against data not used in the calibration. We see that the ensemble spread is reduced for all methods and years (Fig. 3). This is most significant for the NEE, which started off with larger errors relative to the magnitude of the time series than LE. We see the largest reductions in the RMSD for the BFGS and GA minimisations in the calibration year, where the median of each boxplot reduces by over 80%. When applied over the

evaluation period, the reduction in RMSD is not as severe - especially when considering LE. For NEE, the resulting RMSD for the HM experiment is more consistent between the calibration and evaluation periods, suggesting the more conservative approach has stopped us from overfitting to the calibration year. This consistency is less apparent for LE, but we do still see more overlap between the RMSD for the calibration and evaluation period for HM than the two minimisation methods.





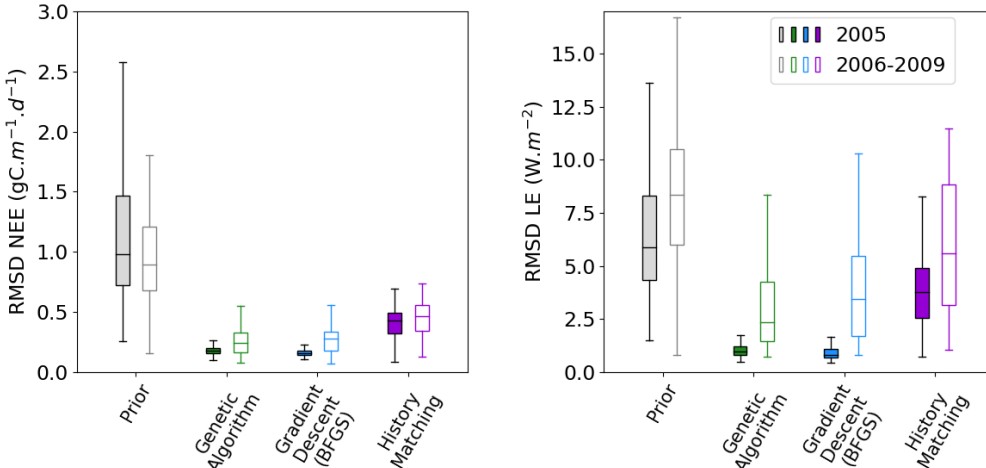

**Figure 3.** Boxplots showing the distribution characteristics of the RMSD of each of the 200 runs for the calibration year (2005, filled boxplots) and the evaluation years (2006-2009, outlined boxplots). The box represents the interquartile range, containing the central 50% of the data. The horizontal line inside the box marks the median. Whiskers extend to the minimum and maximum values within 1.5 times the interquartile range. Full time series corresponding to these plots can be found in Fig. C1.

### 3.1.2 Posterior parameter uncertainty

**Variational data assimilation**

In this next section, we take a closer look at the posterior parameter distributions themselves. As described in Sect. 2.2.2, after minimising the cost function in a VarDA experiment, we usually use information about the curvature of the parameter space to calculate the posterior covariance error matrix $\mathbf{B}_{\mathrm{post}}$ (Eq. 6). Figure 4a shows results from the single GA and BFGS optimisation experiments (i.e., the optimisation with the randomly generated prior closest to the true values). For both optimi-

sations, the reduction in parametric uncertainty is quite severe for all parameters, and for half the parameters, the true value does not fall in the posterior distribution. The differences observed between the optimisations are mainly because we did not converge to the same $\mathbf{x}_{\mathrm{post}}$. The two most sensitive parameters ($\mathrm{VC}_{\mathrm{max}}$ and $\mathrm{Q}_{10}$) have the lowest posterior uncertainty. While still tightly constrained, SLA has the largest posterior uncertainty after both optimisations. After BFGS, $\mathrm{Evap}_{\mathrm{res}}$ and $\mathrm{Root}_{\mathrm{prof}}$ have a larger uncertainty than after the GA optimisation. This suggests that the minimum found from the GA optimisation is

more constrained than the minimum found from the BFGS optimisation.

In Fig. 4b, we consider the impact each of the two fluxes has on the parameter posterior distributions (following the decomposition in Eq. 8). Although the decomposition is shown for the BFGS optimisation, the GA optimisation gives similar results. The results show that the full posterior distribution of the different parameters is the intersection between the posterior distribution of each flux. This is most clearly illustrated by parameter $\mathrm{Q}_{10}$. This parameter is highly constrained after the optimisation

for NEE flux. In contrast, this parameter does not impact the modelled LE and, therefore, is not constrained by this flux. As such, $\mathrm{Q}_{10}$ can take any value for this LE, and so the distribution spans the whole range. Therefore, when accounting for both



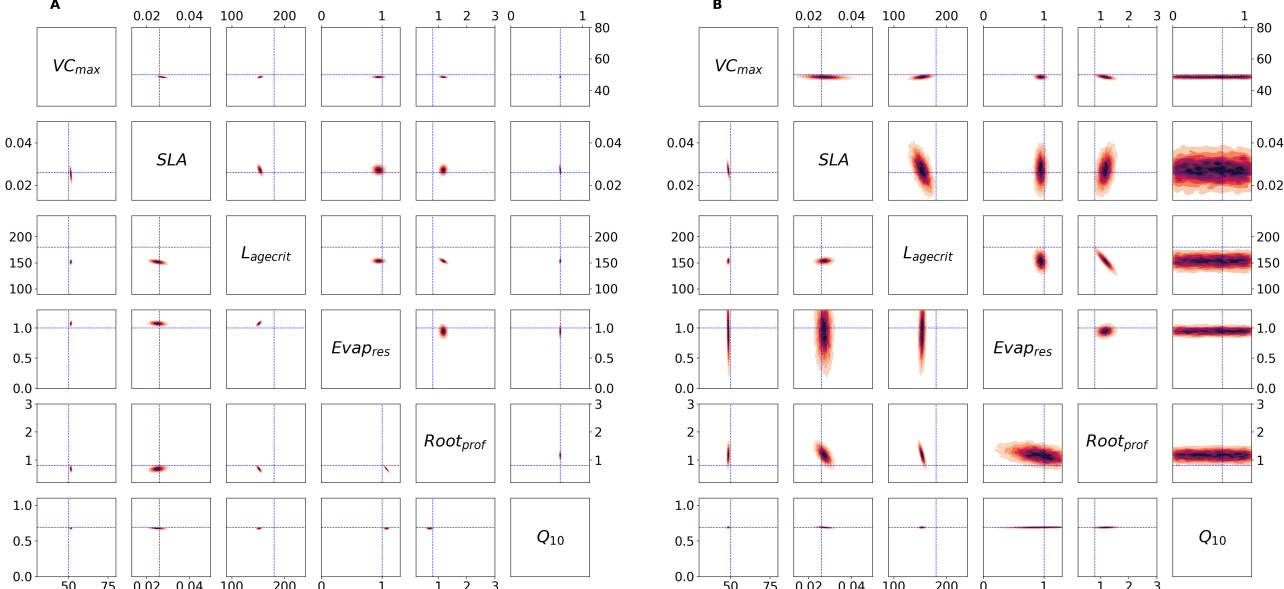

**Figure 4.** Posterior distributions obtained using $\mathbf{B}_{\text{post}}$ (Eq. 6) shown by kernel density estimation plots. Each sub-box is a 2D representation of parameter space showing the density for each pair of parameters, with darker regions signifying areas with higher data density. The true parameter values are shown in blue. a) full $\mathbf{B}_{\text{post}}$ for the GA optimisation (bottom triangle) and the BFGS optimisation (top triangle), b) $\mathbf{B}_{\text{post}}$ decomposition (Eq. 8) for the BFGS optimisation with NEE (bottom triangle) and LE (top triangle).

fluxes, the posterior distribution for $Q_{10}$ matches the NEE constraint. We can further interpret the information in Fig. 4b as the flux sensitivity to each parameter, with constrained parameters being the most sensitive and unconstrained the least sensitive. From this we see that in addition to $Q_{10}$, NEE is more sensitive to SLA than LE, and to $L_{\text{agecrit}}$, although to a lesser extent. LE is

more sensitive to $Evap_{\text{res}}$. Both fluxes give similar constraints on $Root_{\text{prof}}$. These results are consistent with our understanding of the model and the impacts of the different parameters.

Although this decomposition is very informative, it does not explain why the true values do not always fall within the total posterior distribution. There are several reasons why this might be the case, including the two key assumptions made when calculating $\mathbf{B}_{\text{post}}$. First, we assume that we have found the global minimum, and, second, we assume linearity of the model in

the vicinity of the solution, resulting in a Gaussian posterior distribution. This means the $\mathbf{B}_{\text{post}}$ method is unable to take into account any non-Gaussian uncertainty.

We can use the stochastic experiments to bypass these assumptions. Unlike the $\mathbf{B}_{\text{post}}$ method, we no longer have a Gaussian assumption on the posterior uncertainty, allowing us to find non-Gaussian distributions. Furthermore, we have an ensemble of posterior parameters, so the assumption of being at the global minimum is less important. Figure 5 shows the $\mathbf{x}_{\text{post}}$ ensemble

obtained after 200 optimisations, using both minimisation techniques. We see immediately that allowing for non-Gaussian





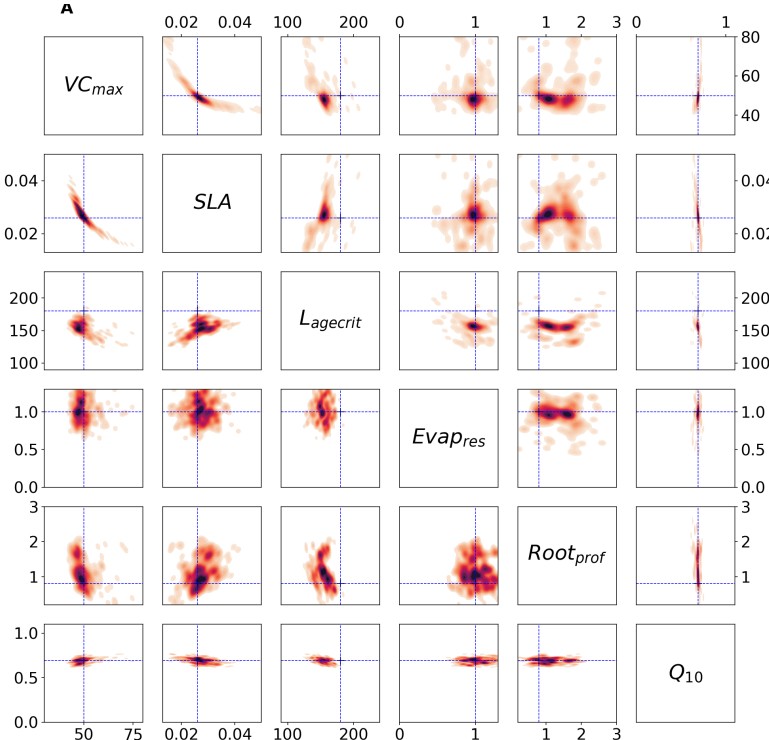

**Figure 5.** Posterior distributions obtained from the stochastic experiment shown by kernel density estimation plots. Each sub-box is a 2D representation of parameter space showing the distribution of the 200 different $\mathbf{x}_{post}$ found with GA (bottom) and BFGS (top), with darker regions signifying areas with higher data density. The true values are shown in blue.

posterior distribution reveals more information about the parameters. For example, a clear relationship between $VC_{max}$ and SLA is found by both minimisation techniques. There is a trade-off between the two parameters - if the leaf has a small surface area (SLA), then the leaf's capacity to capture carbon (through $VC_{max}$) is increased. We further obtain a two-peaked posterior distribution for $Root_{prof}$ (clearest in the BFGS experiment). This parameter defines the depth above which $\sim 65\%$ of roots are

stored. The double peak suggests that either most of the roots is stored above 1.5m, and the trees will primarily get water from the subsurface, or the roots grow deeper to access water down the soil column. Both options would result in the trees having the same water availability. For $Evap_{resp}$, both minimisation techniques remove the possibility of low values and for $L_{agecrit}$, the posterior distribution is centred on the range. We again see that $Q_{10}$ is the most constrained parameter.

     Unlike the $\mathbf{B}_{post}$ experiments, here the true values are contained within the posterior distributions (although $L_{agecrit}$ is found

at the very edge of the distribution, more apparent in the 1D distribution shown in Fig. C2). The distribution of solutions is similar for BFGS and GA. Nevertheless, the GA distributions are slightly tighter and more dense. This is because GA is a global search algorithm and, therefore, less likely to get stuck in local minima. The fact that we still get a large variation in





solutions, while still obtaining a similar fit to the model in Fig. 2, further highlights our problem of equifinality.

## History matching

To directly compare HM to the VarDA approach, in Fig. 6, we use the RMSD between the observations and the model output for both NEE and LE as metrics. Already in the first wave, the $\mathcal{X}_{\mathrm{NROY}}$ reduces by over 80% (Table B1). This is further reduced to less than 10% remaining by the end of the tenth wave. Furthermore, the true parameter values exist in $\mathcal{X}_{\mathrm{NROY}}$. We can also see some of the same patterns we were starting to observe in Figs 4 and 5. Most notable are the relationship between $VC_{\mathrm{max}}$

and SLA and the strong constraint on $Q_{10}$, where values below 0.45 of this parameter are ruled out. Similarly, values of $L_{\mathrm{agecrit}}$ below 124 are ruled out. In contrast, $Evap_{\mathrm{res}}$ and $Root_{\mathrm{prof}}$ are not constrained at all by this experiment; there is not enough information to rule out any values. These two parameters impact LE, specifically its slope in spring and autumn.

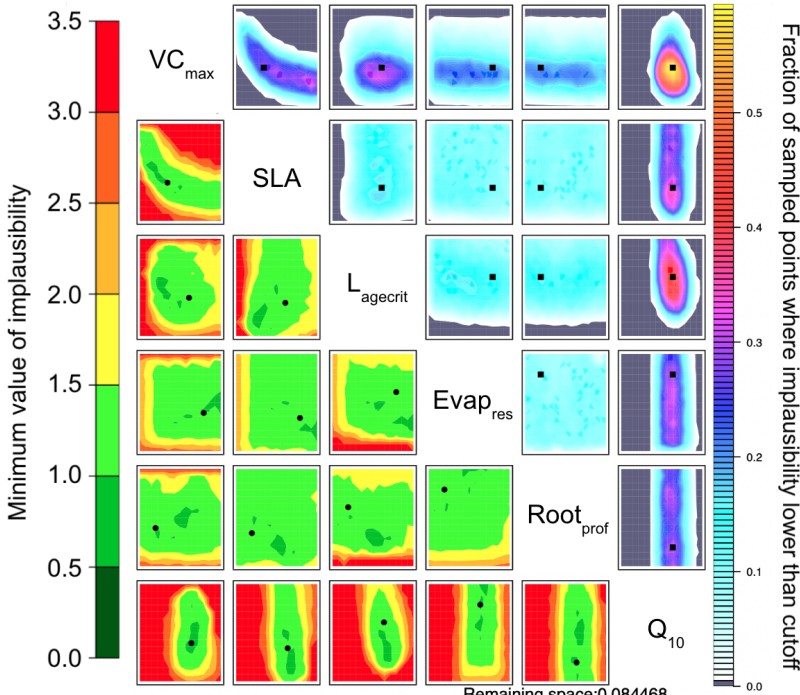

**Figure 6.** NROY density plots (upper triangle) and minimum implausibility plots (lower triangle) from the HM experiment using NEE and LE RMSD as metrics after ten waves. NROY densities (or optical depth) represent the fraction of points with implausibility smaller than the cutoff $a$ (here a value of 3) using the colour bar on the right, with grey regions indicating completely ruled-out areas. This fraction is obtained by fixing the two parameters given on the main diagonal at values of the $x$-axis and $y$-axis of the plotted location and searching the other dimensions of the parameter space. Minimum implausibilities represent the smallest implausibilities found when all the parameters are varied except those used as $x$- and $y$-axes. These plots are oriented the same way as those on the upper triangle to ease visual comparison. True parameter values are shown in black: square on the NROY density plots and circle on the minimum implausibility plots.





### 3.1.3   Computational cost

The computational cost of calibration algorithms is primarily determined by the number of parameters and the time it takes
to perform a single model run. In this study, we test the different methods over a single pixel for a single year, which only
takes seconds to run, meaning the ensembles needed for each algorithm were not too costly to generate. However, in practice, a
single model run can be costly - especially when running the model over a large area or coupled with an atmospheric transport
model. Table 3 shows the number of simulations needed for each algorithm. Note that the extra computation time needed to
construct the emulators and use these to sample from NROY space is marginal in comparison.

**Table 3.** Number of model runs needed in each algorithm for $p$ parameters. Note that the BFGS and GA algorithms were run 200 times for
the stochastic experiments.

|  | Formula | Terms | In this study | Total number of runs |
|---|---|---|---|---|
| BFGS (single run) | $n_{\text{iter}} * (p + 1)$ | $n_{\text{iter}}$ = number of iterations | $n_{\text{iter}} = 25$, $p = 6$ | 175 |
| GA (single run) | $n_{\text{iter}} * g$ | $n_{\text{iter}}$ = number of iterations, $g$ = population size | $n_{\text{iter}} = 25$, $g = 24$ | 600 |
| Bpost | $p + 1$ |  | $p = 6$ | 7 |
| HM | $n_{\text{wave}} * (10 * p)$ | $n_{\text{wave}}$ = number of waves | $n_{\text{iter}} = 10$, $p = 6$ | 600 |

These values represent ball-park figures - a maximum iteration of 25 was used for consistency but the system often converged
sooner (e.g., approximately ten iterations for each BFGS run). Similarly, we used ten waves for HM, but after the 5th wave,
the improvements were marginal (Table B1). Overall, a single BFGS optimisation (i.e., gradient descent) remains the fastest
method. However, it is also the one that is most likely to get stuck in local minima. HM is comparable to a single GA run
in terms of the number of simulations needed. However, we have seen that a single GA run is not enough in this example to
quantify the posterior parameter space fully. Instead, multiple GA optimisations are preferable, which is extremely costly.

### 3.2   Implementing process-oriented metrics

One of the strengths of HM is that we can easily apply different metrics. Indeed, using RMSD is often discouraged since it
is usually associated to a small signal-to-noise ratio. Furthermore, the implausibility (Eq. 10) is already similar to root-mean-
square error (Couvreux et al., 2021). In this section, we consider additional metrics to highlight the power of HM. To select
informative metrics, it can be helpful to identify specific features we want to constrain. For example, for both the NEE and
LE fluxes, we are looking at a seasonal cycle. As such, we expect NEE to have a global sink (i.e., maximum carbon uptake)
and LE to have a global peak (i.e., maximum evapotranspiration) in summer. As well as constraining the magnitude of these
turning points, we might also want to consider constraining when they occur or the rate of change leading to and from them (i.e.
the gradient of slopes). We saw that with the RMSD, we were unable to constrain the $Evap_{res}$ and $Root_{prof}$ parameters. These
parameters impact the slopes of the LE seasonal curve in spring and autumn, respectively, so focussing on these gradients
may help better inform on these parameters. Similarly, $L_{agecrit}$ impacts senescence, so the slope in Autumn is of particular
interest. We also know that in winter, there will be little to no photosynthesis (since we are considering a deciduous site).





Similarly, we expect low rates of terrestrial ecosystem respiration during these months and, therefore, can constrain NEE in
winter. This is similar to constraining the initial carbon pools in the model. In Fig. 7, we consider four of these metrics: i)
Min/Max of the seasonal cycle (sink for NEE, peak for LE), ii) the slope during spring (taken as the difference between April
and February monthly means) and iii) the slope during the senescence period (taken as the difference between September and
August monthly means), and iv) initial carbon stocks (NEE only).

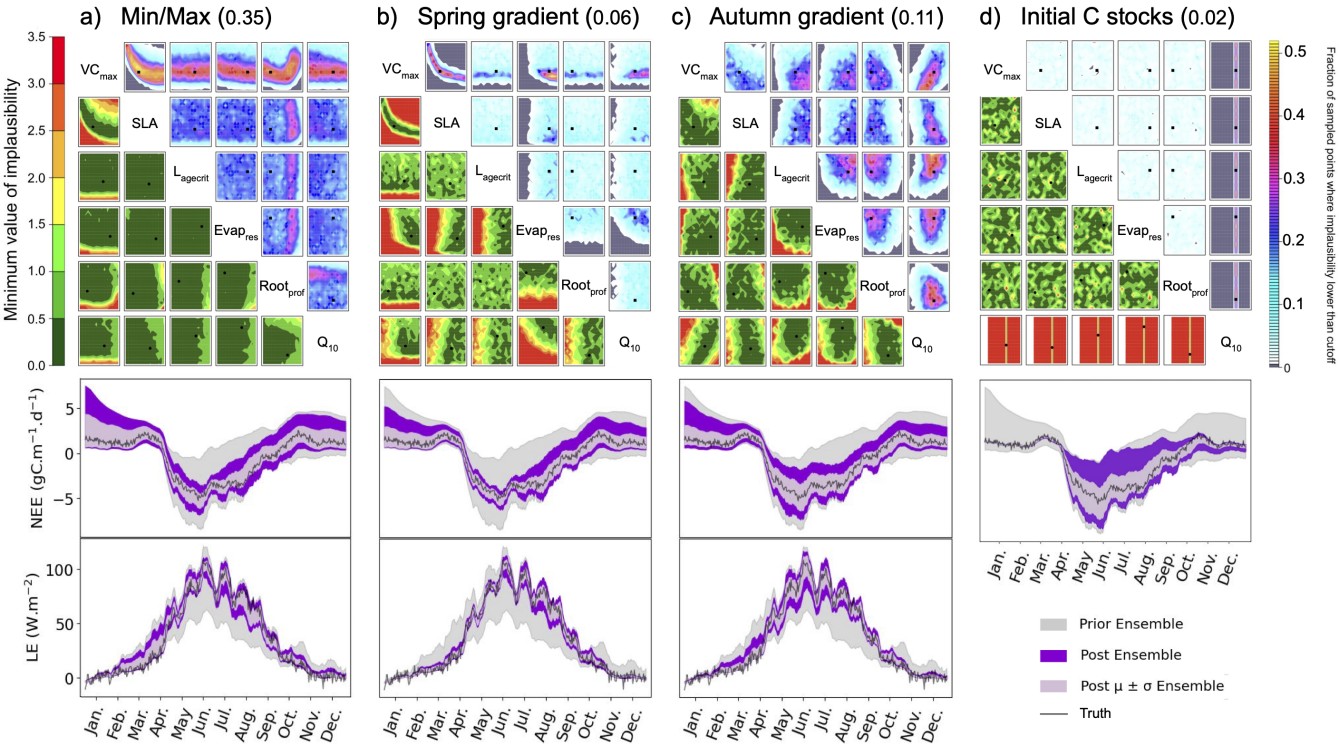

**Figure 7.** History matching experiments considering different individual metrics. NROY density plots are shown above (see Fig. 6 for how
to interpret the figure) with the proportion of remaining space after ten waves in the title brackets. The time series plots shown below each
density plot illustrate the ensemble spread generated running the model with 200 parameter sets sampled from each $\mathcal{X}_{\mathrm{NROY}}$.

In Fig. 7a, we consider how the minimum/maximum of the seasonal cycle can be used to constrain the parameters. We
again highlight the $VC_{max}$-SLA relationship. Another relationship found is between $VC_{max}$ and $Root_{prof}$. Although this metric
cannot be used to rule out unlikely values of $Root_{prof}$, there is a denser fraction of likely points falling around 2.3. Note this
is not the true value of the parameter but is closer to the value of the second minimum found in the stochastic experiments.
When considering $VC_{max}$, $VC_{max}$ is constrained to 50 when $Root_{prof}$ is small. However, when $Root_{prof}$ is large, i.e. the roots can
access water further down the soil column, $VC_{max}$ is less constrained. This means that when there is more water availability,
the carbon capture capacity of the plant is less important in determining the peak and sink of the NEE and LE seasonal cycles,
respectively. Although 35% of the parameter space is left, we see clearly that this metric is insufficient to constrain the other





parameters. Indeed, the minimum/maximum is not sensitive to $L_{agecrit}$, $Evap_{res}$ and $Q_{10}$. When considering the time series for this experiment, the fit is not too dissimilar from Fig. 2, when using the RMSD as the metric. However, we see here that winter behaviour is not constrained. This is especially true for the NEE time series, we do not reduce the spread at the beginning and end of the year.

Figures 7b and c use the slope in spring and autumn, respectively. For the spring slope metric, we again pick out the relationship between $VC_{max}$ and SLA. This is even sharper than in the RMSD and minimum/maximum cases. We also notice a relationship between $Evap_{res}$ and $Q_{10}$, with values in the lower-left-hand corner of the space ruled out. Similarly, low values of $Evap_{res}$ are ruled out using this metric. When considering the time series, we clearly reduce the spread of the ensemble in spring for the LE. We also reduce for NEE, however, this is less obvious since the prior ensemble spread was already quite

narrow. For the autumn metric, we start to constrain $L_{agecrit}$ and $Root_{prof}$, and SLA to a lesser extent. These are parameters that we did not constrain using the other metrics tested. Changes to ensemble spread in the time series are similar to when the RMSD was used as the metric, although a little more marked during September of the LE time series.

The final metric considered in Fig. 7d considers constraining the initial carbon stocks. We clearly see that this metric is completely controlled by a single parameter, $Q_{10}$. This parameter throughout has been the most constrained, and here we see

why. It directly impacts the spread of NEE at the beginning and end of the time series. For the rest of the time series, this parameter has no impact - the ensemble spread in summer is at its maximum width.

These examples illustrate clearly how we can use individual metrics to target different parts of the seasonal cycle. The next step is to combine them in one experiment. In Fig. 8, we combine these five metrics (RMSD, amplitude, spring slope, autumn slope, initial carbon stocks) to have a total of nine constraints (each of the metrics are applied to both NEE and LE, except for

the initial carbon stock metric which is only applied to NEE, see Table 2).

Using these multiple metrics, the $\mathcal{X}_{NROY}$ is reduced to 0.01% of its original size. We see that all of the parameters are constrained, with the true points still contained in the NROY space. Indeed, the true values lie in the space where the minimum implausibility is at most 1.5. We would still recover these points if the cutoff was decreased from its current value of 3. The relationship between $VC_{max}$ and SLA is kept, and we can pick out the true value of $Q_{10}$. By comparing to Fig. 6, we can see

that combining different metrics has helped reduce the space to a much greater extent by constraining the another parameters that were not constrained by solely relying on the RMSD. When considering the time series, we see that the posterior ensemble of model runs tightly fit the observations - especially at the beginning of the year. The ensemble spread is more reduced than when we only used the RMSD (Fig. 2), but with the same consistency between the calibration year and the rest of the time series, especially for NEE.

By using these physical metrics and targeting different parts of the seasonal cycle, we are able to better fit the data and gain an understanding about the parameters. We still have some variability in late Autumn, with the model runs tending to underestimate NEE. Currently, we only target the slope between September and August monthly means to capture the leaf senescence. If we further included a metric in October or November, we may be able to constrain the time series further. Ideally, this would be based on some physical process to understand which parameters would be the most impacted. We also





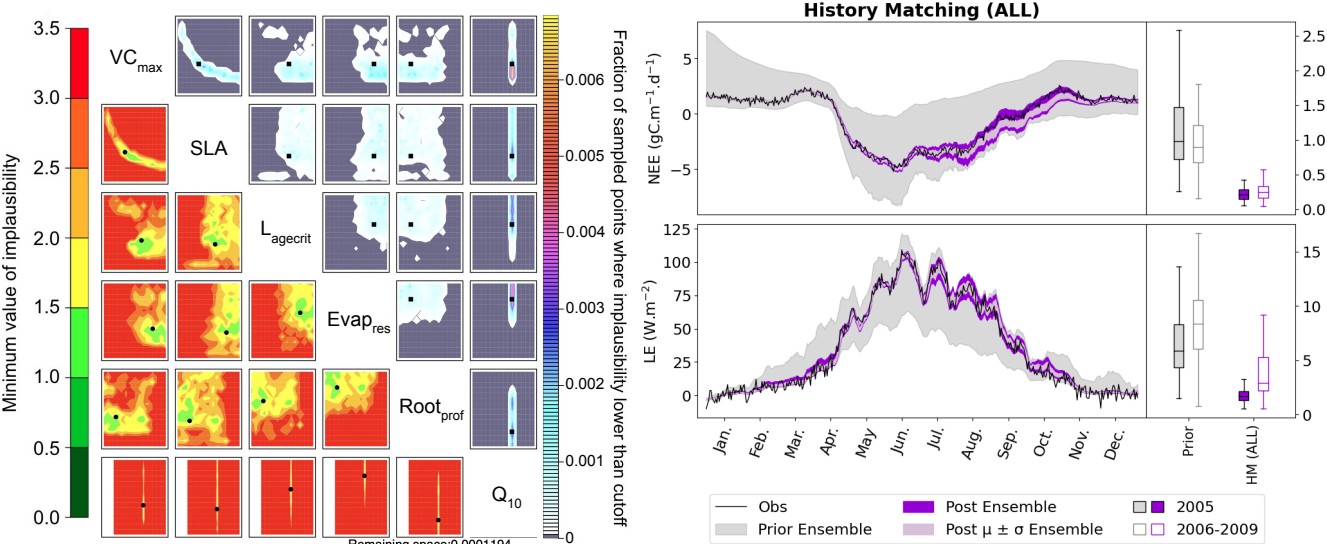

**Figure 8.** History matching experiment considering all metrics listed in Table 2. Results from the third wave are shown with NROY density plot on the left (see Fig. 6 for how to interpret the figure). The time series shown on the right depicts the spread of 200 ensemble runs generated from points sampled from the $\mathcal{X}_{\mathrm{NROY}}$ (shown in dark purple) with the mean and standard derivation of this spread shown in light purple. The boxplots in the right-hand panel show the distribution characteristics of the RMSD of each of the 200 runs for the calibration year (2005, filled boxplot) and the evaluation years (2006-2009, outlined boxplot).

expect targeting the variability in the summer months would help constrain LE during July and August. This would most likely target $\mathrm{Root}_{\mathrm{prof}}$ and $\mathrm{Evap}_{\mathrm{res}}$ as variability during these drier months will impact the amount of water in the soil.

## 4    Discussion

This study provides a good introduction to how HM can constrain the carbon and water cycles in land surface models. Nevertheless, it is not without limitations. In the first part of the study, we compare it to VarDA typically used for parameter
estimation in land surface models, and consider two different minimisation algorithms to reduce the cost function. However, properly comparing VarDA and HM is hard since they are both designed with different goals. This means that discussions around computational cost are nuanced - there are no comparable convergence criteria. Furthermore, for each algorithm, there is a trade-off in computational time and efficiency. For example, we can use more model runs to fit emulators in HM or increase the population size in GA.

In this study we use gradient-descent BFGS and random-search GA to find the optimum and then the gradient information at the optimum to find the posterior uncertainty (through $\mathbf{B}_{\mathrm{post}}$, Eq. 6). We do acknowledge that there are other methods we can use to minimise the cost function - ones where the ensembles can be used to infer the posterior distributions (e.g., Markov-Chain Monte-Carlo, Geyer (1992)). However, these are extremely costly and therefore outside the scope of this study. Furthermore, we




wanted to use the method currently used in ORCHIDAS and, therefore, where we have the most understanding and experience.

Finally, whilst we do not expect it to have a huge impact on the results, without the adjoint of the ORCHIDEE model, $\mathbf{B}_{\text{post}}$ only approximates the curvature of parameter space, and therefore, the posterior parameter distributions found with this matrix are not accurate. Maintaining the adjoint of a complex model like ORCHIDEE is very costly, given the model's evolving nature. Therefore, the fact that HM does not need the adjoint in its calculation of posterior distributions is an advantage.

In the second part of the study, we look at using different metrics for training emulators to use in HM. Although using

different metrics is also possible in VarDA, this is trickier since the choices must be made before a costly calibration, for example, which metrics to use and how to weight them. With HM, we can run an ensemble first and test different metrics on the ensemble. Adding new metrics as the waves progress is also possible instead of restarting the whole optimisation procedure. In this study, we chose a few illustrative metrics based on our understanding of the physical processes driving the model and the results of the one-at-a-time sensitivity analysis (Fig. A1). Choosing the best, most suitable metrics could be the subject

of a whole separate study. Furthermore, increasing the objectivity of the tuning procedure will allow the climate modelling community to more meaningfully share insight and expertise with each other, thereby increasing our understanding of the integrated climate system. In addition to performance-based metrics (e.g., RMSD, skill scores), physical-based metrics allow us to target different parts of the model. However, these require an understanding of the different processes involved. While here we focused specifically on parts of the seasonal cycle, there may be other more complex metrics to consider, for example,

the timing of the leaf-area index reaching a certain threshold. Furthermore, we can use techniques such as principal component analysis to help reduce the dimensionality of the problem (Lguensat et al., 2023)

Nevertheless, HM does come with its own challenges. It is, in a way, much more involved and requires a level of understanding to ensure the emulators are properly constructed and applied. There are also a number of subjective choices that can be made during an HM experiment, such as changing the cutoff ($a$ in Eq. 11) if the emulator is believed to be good enough, which

requires some level of understanding beyond using the method as a black box. When combining a large number of the metrics, it is also possible to expand Eq. 11 to allow for some additional flexibility (Couvreux et al., 2021). For different metrics $m_i$, $\mathcal{X}_{\text{NROY}}$ is the intersection of the $\mathcal{X}_{\text{NROY}m_i}$ associated with each metric:

$$\mathcal{X}_{\text{NROY}} = \bigcap_i \mathcal{X}_{\text{NROY}m_i} = \left\{ \mathbf{x} \in \mathcal{X} : \#\{\mathbf{x} \in \mathcal{X} : \mathcal{I}_{m_i}(\mathbf{x}) > a\} \leq \tau \right\}. \tag{13}$$

where $\#$ denotes the number of metrics that satisfy the condition in the bracket, and $\tau$ is the number of metrics for which the

model is allowed to be far away from the target. Throughout, we set $\tau = 0$, meaning that each model run must satisfy all the metrics. However, when combining many metrics, this value can be increased to stop us from accidentally ruling out good runs before the emulators are properly trained. Although not necessary in our test case, this $\tau$ can be an important consideration.

The example we test here is illustrative but very simple. Several studies have shown moving from a single site setup to multisite optimisations (i.e., finding one common set of parameter for multiple sites) results in more robust optimisation, less

likely to get stuck in local minima (Kuppel et al., 2012; Raoult et al., 2016; Bastrikov et al., 2018). This is because the multiple constraints from the different sites smooth out parameter space. In addition to multisite optimisations, the true strength of HM and its use of emulators will become more apparent when we move to optimise larger regions and more costly processes (e.g.,





spin up). Finally, although the twin experiment is very informative, the next step will be to use real world data to assess the full potential of HM.

### 4.1    Future avenues

HM is a promising approach that may prove invaluable in future land surface model calibration. One of the key challenges in land surface model calibration is multi-data stream optimisations. Ideally, we would perform simultaneous calibrations, where all the information is ingested in one go. However, this is not always practical. There may be some technical constraints, for example, computational capabilities. We may also want to assimilate a newly acquired data stream without redoing the whole calibration process. The alternative is to perform calibrations in a stepwise manner by treating the data sequentially. If dealt with properly, the stepwise approach is mathematically equivalent to the simultaneous one (MacBean et al., 2016; Peylin et al., 2016). However, this means propagating the full parameter error covariance matrix between each step, which can be hard to estimate properly. This is where HM becomes particularly attractive. The NROY space contains all the information about the parameter errors, so this information is not lost between steps. Furthermore, HM's iterative nature means we can add different data when and as they become available. Finally, HM's conservative nature means we are less likely to overfit to a particular data stream or indeed the particularities of given experiments, as shown here.

Although HM does not provide an optimal set of parameters at the end, this is not necessarily an issue. As computers become more powerful, we can run land surface models as ensembles instead of a single realisation of the model, allowing us to obtain rigorously the uncertainty of the model prediction. Indeed, as a climate community, we should be moving towards using data-constrained ensembles instead of a single realisation (Hourdin et al., 2023). HM would allow us to generate such ensembles to be used, for example, the Coupled Model Intercomparison Project. Alternatively, we could use HM for pre-calibration (Edwards et al., 2011), i.e., reducing parameter space before performing data assimilation. The $\mathcal{X}_{\mathrm{NROY}}$ could further help us define the off-diagonal elements of the $\mathbf{B}$ matrix in Eq. 5. We also note that one wave of history matching is as costly as a Morris sensitivity analysis commonly used to assess parameter importance (Sect. A) but much more informative.

The true strength of HM is its ability to identify structural errors. Although we can also use VarDA to identify structural errors, HM's more conservative approach to parameter rejection can more easily help identify when the model is clearly wrong. Furthermore, once an ensemble is generated, applying different metrics to test different model sensitivities is easy instead of performing a full optimisation, as would be needed in VarDA. It is common to add model complexity to models to address structural changes without first checking that the errors truly represent structural deficiencies and are not simply an artefact of poor model tuning (Williamson et al., 2015). Through HM, we can easily test for structural errors and see whether it is possible to match observations given the current model structure.

## 5    Conclusions

Using a twin experiment (i.e., with known posterior parameter values), we first compared the posterior parameter distributions found after variational data assimilation (VarDA) experiments to those found after a history matching (HM) experiment. We



found that the Gaussian hypothesis used to calculate the posterior uncertainty after the VarDA experiments was too strong. The posterior parameter distributions did not reflect the full equifinality of parameter space, nor the non-linear behaviours and relationships of the different parameters. Furthermore, the true parameter values were not contained in the posterior distribution for half the parameters. When performing multiple cost function minimisations starting from 200 different random priors, we achieved a better exploration of parameter space. These experiments were much more computationally costly but started to

reveal relationships between parameters and contained the true parameter values in the posterior distribution. Similarly, while not constraining all the parameters, the posterior distributions from the HM experiment also contained the true parameters and maintained non-linear relationships between parameters. Furthermore, the model ensemble found by sampling the not ruled-out yet space fitted the observed time series reasonably well.

In the first part, we used the root-mean-squared difference as the target metric for HM to directly compare to the cost function

used in the VarDA experiments. In the second part, we showed HM's versatility in using different metrics to target different parts of the seasonal cycle. This allowed for all the parameters to be better constrained and the posterior model ensemble to tightly fit the observations. We showed that instead of using a single cost function, multiple process-based metrics improved the calibration, and enhanced our understanding of the model processes.

Although this paper only considers a simple exploration of the HM methodology, its strong potential for land-surface model

calibration is clear. It will allow us to constrain multiple data streams, better targeting individual processes. Overall reductions in parameter uncertainty will lead to more accurate projections of the land surface, enhancing our understanding of terrestrial behaviour under climate change allowing us to better plan for the future.

*Code and data availability.* The source code for the ORCHIDEE version used in this model is freely available online via the following address: https://doi.org/10.14768/d64cfc44-08b7-4384-aace-52e273685c09 The ORCHIDAS HM code and data used in this paper are avail-

able on a GitHub repository: https://doi.org/10.5281/zenodo.10592299. The FR-Fon Fluxnet meteorology data can be directly downloaded from the FLUXNET2015 database after registration: https://fluxnet.org/data/fluxnet2015-dataset/.

## Appendix A: Parameter sensitivities

One of the primary uses of HM is to test the sensitivity of the model outputs to the parameter uncertainty. Nevertheless, we ran a couple of simple tests in advance to get a sense of the different parameter sensitivities. These more closely resemble the

tests land surface modellers perform when doing simple manual tuning and are not as rigorous as HM. However, they remain common - especially since they can be used to pre-select the key parameters for these more robust but costly methods.

The first is a simple, one-factor-at-a-time parameter perturbation experiment (Fig. A1). Although this does not account for interactions between parameters, it can help understand some of the direct impacts of the different parameters. This cost of the method is determined by the number of ensembles used for each parameter, here 50. The second is a Morris sensitivity analysis

(Morris, 1991; Campolongo et al., 2007)(Fig. A2), which is effective with relatively few model runs compared to other more





sophisticated methods (e.g., Sobol', Sobol (2001)). Indeed, a standard Morris experiment equates to the computational cost of doing one HM wave (i.e., 10 times the number of parameters forward runs of the model). Using an ensemble of parameter values, the Morris method determines incremental ratios, known as 'elementary effects'. These effects are determined by sequentially modifying individual parameters across multiple trajectories within the parameter space. The mean ($\mu$) and standard

deviation ($\sigma$) of the differences in model outputs for all the trajectories are calculated. This global method determines which parameters have a negligible impact on the model and which have linear and non-linear effects. The results of this method are qualitative and serve to rank the parameters by their significance. To evaluate the results, we consider the normalised means, obtained by dividing each value by the $\mu$ of the most sensitive parameter. As such, the values fall within the range of 0 and 1, with 1 indicating the most sensitive parameters and 0 indicating parameters with no sensitivity.

Both methods clearly show that NEE and LE are highly sensitive to $VC_{max}$ and SLA. In Fig. A1, we see that these parameters directly impact the amplitude of the seasonal cycle. NEE is most sensitive to $Q_{10}$ (the parameter determining the temperature dependence of heterotrophic respiration), which we see impacts the respiration (TER) component of this flux. This parameter has no impact on LE. After $VC_{max}$ and SLA, LE is most sensitive to $Evap_{res}$, which controls bare soil resistance to evapotranspiration. This parameter impacts the slope in spring - controlling how much water is in the soil before the leaves start to grow

at this deciduous site. Both fluxes are sensitive to $L_{agecrit}$ in autumn, this parameter impacts the age of leaves and therefore senescence. Finally, while still showing some sensitivity, the fluxes are least sensitive to $Root_{prof}$. This parameter controls root depth and seems to have the most impact in summer when the months will be warmer.

While the one-factor-at-a-time experiment seems more informative here, we must remember that it does not account for the interactions between the parameters. This can be dangerous because it is possible that changing different combinations of the

parameters may impact the processes differently. This will be crucial in more complex cases. While the Morris experiment does account for interactions, it is not possible to disassociate from non-linear effects. Furthermore, although Morris is less costly, little information beyond simple rankings can be gleaned.

## Appendix B: Emulator quality

We ran leave-one-out cross-validations to validate the emulators at each wave of the HM procedure. Each point from the design

set was retained for validation and the emulator refitted. We then test if that point lies within the 95% confidence interval of the refitted emulator - if 95% ratio of the left-out points should lie inside the confidence intervals, then the emulator is deemed good. While it is an ideal case, we consider in this work that the emulator is good if the ratio is at least over 90%.

To illustrate this, Fig. B1 shows the diagnostic plots for the first and last wave of the HM RMSD experiment, and Table B1 shows the leave-one-out diagnostics at each wave. The emulator represents the model well for both the NEE and LE cases

with small error bars and predictions close to true values. We obtain larger error bars for NEE compared to LE (especially in later waves) due to the more sophisticated behaviour of the model for the NEE case. In Table B1, we see that both the average error and variance decrease with successive waves, showing our emulators are becoming more accurate, and in each case, the





**Figure A1.** Prelimarily experiment demonstrating the individual sensitivity of each parameter. For each row, 50 ensembles of a given parameter are shown for NEE and LE, as well as NEE's components - gross primary production (GPP) and terrestrial ecosystem respiration (TER). Runs are coloured by the parameter values used: low values in blue, passing through yellow and high values in red.



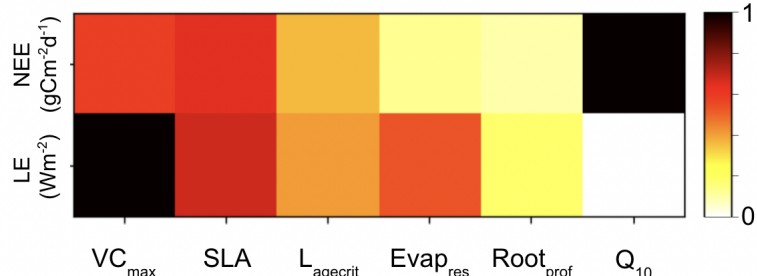

**Figure A2.** Heatmap showing the relative sensitivity of each parameter for NEE (top row) and LE (bottom row ). Morris scores are normalised by the highest ranking parameter in each case. Dark squares represent the most sensitive parameters for each output, and light squares represent parameters with little to no sensitivity.

**Table B1.** Information about emulators and the leave-one-out diagnostics at each successive wave. The error column shows the mean emulator error and s.d. column shows the mean emulator std-deviation for each prediction.

| | Fraction of space | NEE | | | LE | | |
|---|---|---|---|---|---|---|---|
| Wave | remaining | Accuracy (%) | Error | S.d. | Accuracy (%) | Error | S.d. |
| 1 | 0.1428 | 92.1 | 0.103 | 0.11 | 92.2 | 1.07 | 1.08 |
| 2 | 0.1172 | 92.9 | 0.067 | 0.065 | 92.9 | 0.496 | 0.444 |
| 3 | 0.1092 | 94.2 | 0.057 | 0.059 | 94.2 | 0.303 | 0.282 |
| 4 | 0.1092 | 92.7 | 0.056 | 0.054 | 94.7 | 0.282 | 0.275 |
| 5 | 0.0948 | 93.6 | 0.037 | 0.035 | 94.6 | 0.186 | 0.175 |
| 6 | 0.0932 | 94.2 | 0.028 | 0.028 | 94.9 | 0.169 | 0.154 |
| 7 | 0.0864 | 94.9 | 0.025 | 0.025 | 95.2 | 0.147 | 0.138 |
| 8 | 0.086 | 94.5 | 0.025 | 0.024 | 95.6 | 0.111 | 0.107 |
| 9 | 0.086 | 94.1 | 0.022 | 0.023 | 95.2 | 0.109 | 0.103 |
| 10 | 0.0845 | 93.5 | 0.021 | 0.021 | 94.8 | 0.084 | 0.084 |

accuracy is above the 90% mark. Overall, we are satisfied with the quality of our emulators. Although not shown, emulators from the other HM experiments give similar results.



**Figure B1.** Leave-One-Out diagnostics plots against each of the parameters for NEE (top row) and LE (bottom row) for the first and last wave of the history matching RMSD experiment. Black points and error bars ($\pm 2$ s.d. prediction intervals) are computed from $\mathrm{E}[H(\mathbf{x})]$ and $\mathrm{Var}[H(\mathbf{x})]$. The true (left out) values are plotted in purple/red if they lie within/outside two standard deviation prediction intervals. The horizontal dashed lines show the observations plus the observed error ($0 \pm 2\sqrt{\mathrm{Var}(\mathbf{e})}$). Note the difference in scale between the plots in Wave 1 and Wave 10.





## Appendix C: Additional figures

Here, we present a few additional figures to help further visualise the results. Figure C1 shows the time series for the additional evaluation years (2006-2009).

**Figure C1.** Full time series of NEE (top) and LE (bottom) for FR-Fon years 2006-2009. For the data assimilation experiments (GA and BFGS), the spread represents the results taken from the stochastic experiments (i.e., from 200 optimisations). For the HM, 200 parameter sets were sampled from the $\mathcal{X}_{\mathrm{NROY}}$ and used to run the model. The prior ensemble, i.e., before any calibration, is shown in grey and the posterior ensemble is shown in dark colours. Boxplots on the right show the distribution characteristics of the RMSD between each of the 200 runs and the observations. The RMSD for the calibration year (2005) is shown by filled boxplots, and the evaluation years (2006-2009) are shown by outlined boxplots. The box represents the interquartile range, containing the central 50% of the data. The horizontal line inside the box marks the median. Whiskers extend to the minimum and maximum values within 1.5 times the interquartile range.



Figure C2 compares the 1-dimensional posterior distributions found by the VarDA stochastic experiments and HM experiments. Remember that this is a 1D representation of multidimensional space and so does not illustrate the relationships between
parameters.

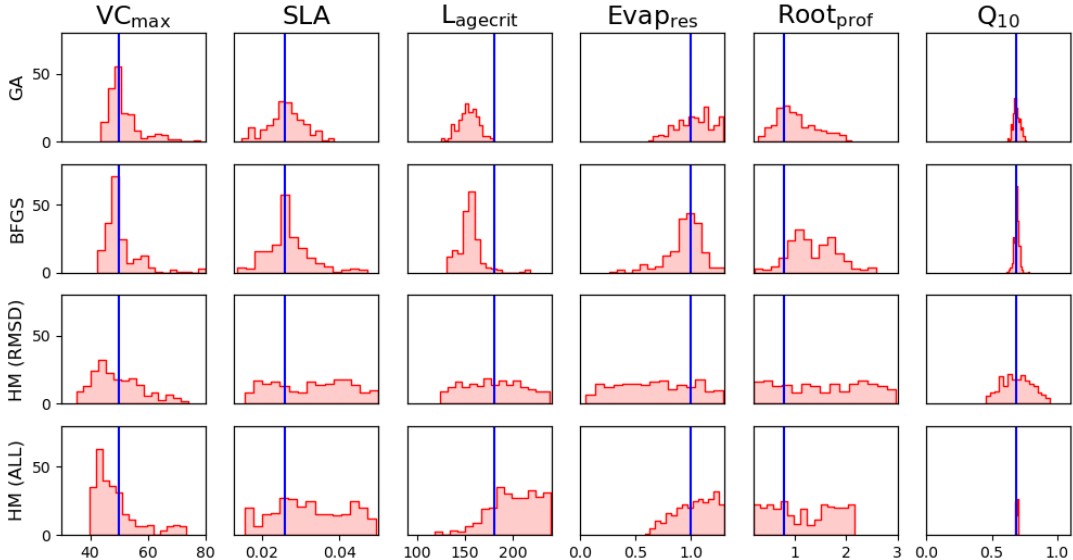

**Figure C2.** 1-dimensional representation of posterior parameter distributions. For the data assimilation experiments (GA and BFGS), the values represent the results from the stochastic experiments (i.e., from 200 optimisations). For the history matching experiments (HM; RMSD for when only the RMSD metric is used, ALL for when all the metrics are applied), 200 parameter sets were sampled from the $\mathcal{X}_{\text{NROY}}$. The true values are shown in blue.

*Author contributions.* NR and PP conceived on the study. SB performed the DA experiments. NR performed the HM experiments. NR and SB helped integrate the HM methodology in the ORCHIDAS system maintained by VB. FH and JS provided expertise in assessing the HM results and the emulator performance. CO provided supervision and DA expertise. All authors contributed to the writing and editing of the manuscript.

*Competing interests.* At least one of the (co-)authors is a member of the editorial board of Geoscientific Model Development

*Acknowledgements.* We would like to acknowledge Maxime Carenso for his help running preliminary experiments during a summer placement. We also acknowledge Daniel Williamson and Victoria Volodina, University of Exeter, for their work integrating the History Matching method into 'HighTunes' for use with the LMDZ model, on which our ORCHIDAS integration was based. We would further like to thank




the core ORCHIDEE team for maintaining the land surface model code and the Fluxnet community for acquiring and sharing eddy covari-

ance data (here from the Fontainebleau-Barbeau site, Delpierre et al. (2016)). This work has received funding from the European Union's Horizon 2020 research and innovation programme under the Marie Skłodowska-Curie grant agreement No 101020076. SB is supported by a scholarship from CNRS under the Melbourne-CNRS joint PhD programe.



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
