# Peer review of "Exploring the Potential of History Matching for Land Surface Model Calibration"

_EGUsphere, 2023_

## Author Comment (AC1)

**Review 2**

This study explores the use of history matching and the not-ruled-out-yet (NROY) parameter space to calibrate land surface model parameters and quantify uncertainty on model outputs. The method is demonstrated in a twin experiment with the ORCHIDEE model (i.e. using model-generated "observations" with known parameters), and compared to parameter optimisation and uncertainty characterisation by a gradient-based method and a global-search method.

The paper is very clearly written, and provides a valuable demonstration of the history matching / NROY method. The comparison with the gradient-based and global search methods is an important part of the paper, as these methods have often been used to optimise parameters and characterise model uncertainty in land surface models. The comparison of the uncertainty range from the ensemble of 200 optimisations with Bpost and with the HM range is also good to see. I like the exploration of different metrics. The results are significant and I recommend publication of the paper with minor revision to address the comments below.

We would like to thank the reviewer for taking the time to read and comment on the manuscript. They make some interesting points, which we have incorporated into the manuscript as best we can.

Specific comments:

Line 11 - "the true parameters are contained in the posterior distribution" - is this guaranteed with history matching, or do you find that it occurs in this example and could there be cases where it doesn't?
The fact that it contains true parameters shows how well HM works. This sentence can fail to be true if the emulators are bad/biased leading one to rule out good parts of space because the emulators are not being a suitable (probabilistic) representation of the model. Furthermore, misspecification of errors can also lead to ruling out the whole space: if Var[f(x)] -> 0 and one had set e.g. Var[e] and Var[eta] too small. However, it is possible to revisit these tolerances to keep something, which should then retain the 'truth' - but only if the emulator is good enough.

Line 73 - VarDA (and the term variational) is used here to include both the gradient-based method and the genetic algorithm. I am used to using the term 'variational' to describe gradient-based methods, in contrast to terms like Monte Carlo, global search, or stochastic to describe a genetic algorithm. I wasn't able to find a definitive definition of variational, and it seems to be used differently by different authors (e.g. Santos et al. (2013, doi: 10.1590/S2179-84512013005000012) contrasting variational methods and genetic algorithms, and Schmehl et al (2011, doi: 10.1007/s00024-011-0385-0) describing a genetic algorithm variational approach). Nonetheless, I think it is worth considering whether a different term would be better to describe the two parameter optimisation methods (e.g. ParOpt for Parameter Optimisation, or ParEst or PE for Parameter Estimation) to avoid any possible confusion.

This is a tricky point since, as the reviewer points out, in the literature, variational can both be interpreted as "minimising the cost using gradient-based methods" and simply "minimising the cost function" (regardless of technique). In all cases, the main focus of the literature is usually on constructing the cost function, and therefore variational is commonly used to refer to the type of cost function used more than the method used in the minimisation. While reviewers suggestions for alternative terms have merit, we feel they are a) too vague, especially given the fact we want optimisation with uncertainties (as described by the 4D-Var equation) and b) one could argue that history matching could fit into the realm of of parameter estimation as the goal is still to find viable parameter sets by ruling out the unlikely ones. We have decided the keep the term "VarDA" but with more transparency on L128 (in accordance with RC1), which hopefully will ease confusion:

**Note that here we use the term variational to describe the form of the cost function minimised. While the classical approach to minimise this function relies on gradient-based methods, in the absence of gradient information, other methods have increasingly been used to find the optimum. This has led perhaps to an abusive use of term "variational", however, we feel here it helps to group, via a common cost function, the two minimisation approaches we wish to compare to the history matching approach.**

Line 156 - this equation assumes $\sigma_i$ is the same error for all observations in each stream

That is correct, the statement has been added to the text.

Line 165 - define E

We have added the following to line 167:

**where E is the expectation and Var is the variance.**

Figure 1 - I don't understand the text between the first pink shape and the first purple shape "For 1st wave $\chi = \chi_{NROY}$". I would understand it if it defined the first wave $\chi_{NROY} = \chi$.

Apologies for the confusion, the reviewer is correct the equation needs to be reversed. The figure has been amended as such.

Line 225 - write 10,000 rather than 1e4

Done

Line 313 - Q10 is the most constrained parameter *relative to the prior range*.

Added

Line 340 - could remind the reader here that 200 GA optimisations were used e.g. "Instead, multiple GA optimisations were preferable (we used 200), which is extremely costly."

Added

Figure 7 - Is Min/Max the quotient of the min and max of the data? Please define exactly what this metric is. What is the number beside the panel caption (i.e. 0.35 beside Min/Max, 0.06 beside Spring gradient etc)? Vertical gray lines could be added to the timeseries for b) at Feb and Apr and c) at Aug and Sep to point out the months used in the metrics. The constraint of initial carbon stocks, is that something that is often observed?

Line 383 - be consistent in using min/max or amplitude to describe that metric.

Amplitude on L383 changed to min/max

Line 383 - Do you need to weight the different metrics when they are combined in HM?

Not in the traditional sense. We have added the following to L385:

> **Note that these metrics are not weighted (in the traditional sense) when combined. Instead, the weighting occurs through the individual errors used to set up the experiments.**

Is there noise added to the observations used for the alternative metrics? In a twin experiment, without noise on the observations, it is easy to see how the parameters could be much better constrained than in a realistic case with actual observations.

Yes, the noise on the observation (described in L200) is applied throughout.

Line 459 - In the context of land surface models, a stepwise approach to separate calibration of the fast and slow processes (as described at line 85) would benefit from this feature of the HM.

Good point, we have added the following to the text:

> Furthermore, HM's iterative nature means we can add different data when and as they become available. **It also lends itself well to a stepwise approach to calibration, allowing us to separately constrain, for example, the fast and slow processes of the model.**

In contrast with the RC1 comment, I do believe that the VarDA part of the paper is important, as it reflects the way parameters are often optimised and uncertainties quantified in land surface models. Personally, I like the style of discussing the meaning of some of the results given in the Results section, rather than leaving all of that discussion to the Discussion section, but I guess that is a matter of style.

Thank you for saying this, we agree that both approaches have their merits. Here we have decided to keep some of the discussion with the results, leaving space in the "Discussion" section to consider the pros and cons of both techniques more broadly.

---

## Author Comment (AC2)

**Review 1**

This is a review for the manuscript "Exploring the Potential of History Matching for Land Surface Model Calibration" submitted to Geoscientific Model Development by Raoult et al. In the work included in the manuscript, the authors examine the benefits of using History Matching (HM) for Land Surface Model (LSM) calibration with the ORCHIDEE LSM model by conducting a twin experiment reflecting a site often used for ORCHIDEE calibration. In addition they compare the HM results with how VarDA calibration performance.

From the HM implementation/experiment part, I thought the manuscript was well done and successful. It is a basic what is being done here, but the paper is open about that and establishes itself more as a foundation work that will be expanded upon in the future. There are some parts where some clarification is needed, but overall I was quite satisfied with the work done and how it is presented here.

We would like to thank the reviewer for taking the time to read through and comment on this manuscript, comments that will strengthen the paper and help clarify key points of the paper.

Which, however, brings us to VarDA comparison as I did not really comprehend the purpose of it. As the manuscript even itself admits in the discussion section, how those two calibration methods function are so fundamentally different that the results really can't be compared effectively. VarDA is ultimately an optimization method, especially when applied in the manner here, and even the uncertainty approximation used in the work is kind of a fix as the approach itself does not actually produce uncertainties. That has been one of the great challenges in using 4D-Var in forecast service and has produced several different workarounds.

So to use VarDA, which essentially was 4D-var here, to estimate parameter values and then compare those with a method that only estimates uncertainties was odd, to be honest.

The history matching vs variational data assimilation comparison was motivated by the fact that VarDA, formulated in this precise manner, is a common method used in land parameter data assimilation, especially when dealing with the carbon cycle; e.g., BETHY (Rayner et al., 2005, Scholze et al., 2007, Knorr et al., 2010, Kaminski et al., 2013), JSBACH (Schürmann et al., 2016; Castro-Morales et al., 2019), JULES (Raoult et al. 2016, Pinnington et al., 2020), ORCHIDEE (Santaren et al. 2007, Verbeek et al., 2011; Kuppel et al., 2012, 2014; etc). We have added more of these examples to the text. Part of the reason is linked to the history of land surface models when their role was primarily acting as a lower boundary to atmospheric models. VarDA has been extensively used with atmospheric models. When calibrating land parameters linked to the carbon cycle, one of the data sources is atmospheric $CO_2$ mole concentration data. To calibrate against this data, we need atmospheric inversion models creating a link with the atmospheric modelling community. As such, early frameworks for carbon land surface model parameter estimation used VarDA at its core (Rayner et al., 2005, Scholze et al., 2007, 2016, and 2019, Kaminski et al., 2012 and 2013; Peylin et al., 2016; Bacour et al., 2023).

Given this rich culture of data assimilation in land surface models, and in particular in ORCHIDEE (see https://orchidas.lsce.ipsl.fr/publications.php for the over 30 studies using this approach), we felt it was therefore necessary to highlight the advantages of using another method, i.e., history matching, compared with what has historically been used. This was also partially motivated as a follow-up of Bastrikov et al. (2018) where BFGS and GA were compared, and therefore, this adds a new method to the repertoire.

Although HM and VarDA differ fundamentally in their approach, they still have a similar objective of finding acceptable parameters. VarDA searches for THE optimal parameter set while HM searches and eliminates all non-suitable parameter sets with the goal of finding possibly a small and reduced parameter space. In practice, HM has often been used historically to calibrate models. Furthermore, we see in this paper that performing VarDA from multiple different priors helps to get an ensemble comparable to one extracted from the NROY. Therefore, by comparing them in this manner, we can illustrate the advantages of using HM and show how HM could complement the methods currently used to optimise these models.

Furthermore, the more commonly used MCMC methods actually produce uncertainty estimates that would be a more apt comparison for the HM results. While the discussion does touch on why MCMC wasn't used here, it should have been done more explicitly earlier and even then it remains questionable. In truth, it almost felt like the author was worried that the paper itself would have been too short or limited without the comparison, but it doesn't really add that much.

MCMC methods are very powerful, providing a full exploration of parameter space and have the potential of providing very detailed information about the posterior distributions from which it samples. However, these methods are extremely costly, sometimes needing the model to be evaluated upwards of $10^4$–$10^7$ times. Furthermore, the Markovian nature of MCMC techniques, which requires the computation to be performed sequentially, means this process cannot be parallelised. As such, MCMC methods have only been applied to computationally inexpensive LSMs or to calibrate isolated processes - ORCHIDEE, unfortunately is an LSM of high-complexity and therefore MCMC is too costly. Only recently, with the help of emulators, has MCMC been used to calibrate LSMs of high complexity (e.g., Fer et al., 2018).

We have added to the introduction (L41):

> As a result, LSMs are also becoming more costly to run. **This also means that traditional Bayesian calibration techniques, such as Markov Chain Monte Carlo, are too costly to use.**

Furthermore, currently, most groups working with complex process-based land surface models (an important community preparing LSM for the next Climate assessment report) are dealing with the same question as touched on in this paper: using the classical 4Dvar approach or using emerging methods based on emulators like the HM one proposed here. Thus we believe the comparison is highly relevant for the global LSM community.

Apologies on the brusqueness of the feedback regarding the comparison, but it is driven by how much I did appreciate the HM part of the work. If this paper was still in the preparation phase, I

would strongly argue removing the comparison, but at this stage that is too drastic an action to take. And while I continue to have my issues with the VarDA part, I do think the rest of the paper is strong enough to be considered for publication.

We do agree that with these technical papers, we do have a tendency to make them long and in future, a more targeted message would be better. Nevertheless, given the current state of land parameter data assimilation, we do feel that this comparison is beneficial to the community (as does Reviewer 2).

In the HM literature, allusions to this comparison have been made (Williamson et al., 2015; Hourdin et al., 2023) but no paper (to our knowledge) presents this kind of experiment. This can partially be explained by the rather different groups of people working on HM and parameter data assimilation. It was important for us to make this potential link between the communities, as these techniques are becoming increasingly multidisciplinary. What's more, MCMC methods are much closer to HM in their conception, and so these techniques are often merged (e.g. Mohamed et al., 2012; Maschio & Schoizer, 2014; Han et al., 2024). Both MCMC and VarDA have been compared to other ensemble methods (e.g., Fairbrain et al., 2013; Bannister, 2016; King et al., 2024). A gap in the literature comparing HM and VarDA was part of the motivation for this study.

Thus my recommendation is to return for major revision with the focus, in addition to the detailed comments later on, being on strengthening the reasoning and expectations for the VarDA comparison both in the introduction and in the methods section.

Below are my line-by-line comments for the manuscript. As a general comment, though, there are several points where the manuscript refers to something as usually or commonly done which I feel should almost all be removed. Just state what is done and why instead of using generic referrals such as that, especially because they are both unnecessary and arguable in some cases.

We again would like to thank the reviewer for their thoroughness in reading the manuscript and for the line-by-line comment, which we reply to below.

Line-by-line comments:

Line 20: "However, despite their increasing complexity…"

This comment is so nit-picky that I feel compelled to apologize for it in advance, but I might argue against the word despite here. While it is undeniable that more complexity has been attempts to add to LSMs, a lot of those processes are intertwined. Hence it would make sense that, at least in the beginning, one would expect to see those major uncertainties remain as they have more room to spread so to speak.

Not a critical comment and feel free to ignore, but still something to potentially rephrase here.

Rephrased as:

However, **in part due to** their increasing complexity (Fisher and Koven, 2020), these models **are** subject to large uncertainties, in terms of missing processes and poorly constrained parameters.

Line 23: "…terrestrial biosphere is becoming a critical scientific priority…"

I would just expand that it is also a policy priority as the results of these models are used to basis for future plans as explained in the example following this part.

We thank the reviewer for this suggestion, we have expanded this line to include this.

Line 28: "DA can be used to improve the initial state of the model and/or the internal model parameters."

This part, along with the following more detailed examples, are a bit misleading in my opinion as DA is more commonly used to continuously update the state variables as new observations become available. While that new estimated state is then used as the basis for the next projection, it also does contain information from the preceding model states, so calling it the initial state is not completely accurate.

Although DA is more classically associated with numerical weather prediction and correction of model trajectories, for LSM, the initial state is a key part of the model trajectory containing information from the preceding model states found during the spin-up. This definition of DA is very common in the LSM literature. We have added a citation (Rayner et al., 2019) to support this definition.

Also on the numerical weather comparison after this part, I again felt it was a bit inaccurate. First of all, in order for the parameter estimation to be even possible, you need known equation to calibrate. So the fact that the equations were known is not an explanation why the parameter estimation isn't the focus. Furthermore, and more importantly, because of the chaotic nature of weather systems, error related to the current state spreads faster and ends up dominating the future projection error. I would argue that is a more central reason why state data assimilation is used and why with LSMs it makes more sense to focus on model parameter uncertainties.

By the way, this part isn't to be nit-picky like my first comment, but reading through this part I felt it asserted confusing things such as what kind of an equation exists as one cannot use those methods if one doesn't have an equation to begin with. So the issue with LSMs, for example, isn't that they rely on empirical equations, but that there are questions how generally applicable a set of parameters for those empirical equations are and in which situations they should be recalibrated.

The reviewer makes a good point and we have rewritten this section as follows:

In numerical weather forecasting,  DA is predominately used to  **update the** state **of the model, due to the chaotic nature of the weather system. These models are primarily used to provide near-real-time forecasts, and errors in the initial state can dominate**

**the error in short-term future projections.** In contrast, in climate studies, we rely less on initial state optimisation and more on parameter calibration, especially for the carbon cycle, where a lot of processes are based on empirical equations **since we tend to be more interested in long-term trends**. **In addition, LSMs use a small number of parameters to represent a large diversity of ecophysiological properties. As such, the calibration or tuning of this parameter has become central to climate modelling. For a long time, it was done by hand, often linked to the subjectivity of the modeller (Hourdin et al., 2017). The emergence of DA techniques for calibrating parameters has made it possible to focus on more objective criteria and account for uncertainties.**

Line 32: "Furthermore, we often rely on variational data assimilation methods..."

I strongly disagree with the implication here that variational methods are the common approach in climate study approaches. They are used, obviously, but MCMC based methods are still much more used at least based on my experience. Which is actually my biggest source of confusion with this paper, something I will delve more deeply later, in that for this particular experiment it feels like a comparison of History Matching to a MCMC approach would have been much more fitting due to the uncertainty aspect.

Additionally, the description of the variational method is a bit odd as while technically correct, in 3D variational assimilation the assimilation is done for each observation moment separately. So in those cases it is not really a time window, at least not in the manner indicated here.

In land surface model parameter data assimilation, variational methods are common, especially when dealing with the carbon cycle (as shown in response to the first comment). Nevertheless, we have expanded the statement to be more precise by explicitly stating this is what is done for LSM of high-complexity:

> "Furthermore, **when dealing with LSM of high-complexity,** we often rely..."

Next, as discussed above, comparisons between History Matching and MCMC are outside the scope of this paper because a) MCMC is often too costly for complex land surface models and, therefore, we have been using different approaches for calibration, b) such a comparison would not have been novel and not of direct use by the global LSM community.

For the final part of this comment, we agree that the current formulation can cause confusion due to the 3DVar setup. Therefore, we have added the following to ensure the reader knows we are talking about 4DVar (as also requested in one of the later comments).

> Furthermore, we often rely on **a 4-dimensional** variational DA (**simply referred to here as** VarDA) **framework**, in which all observations within the assimilation time-window are used to create a cost function which is then minimised.

Line 50: "- for example, the likelihood…"

I was very confused by the claim here. Yes, by its very nature likelihood will generally be univariate and smooth as it is a continuous variable that has only one maxima/minima. However, it does nothing to address the equifinality issue that is a fundamental challenge to calibration in general, emulators included.

Let us say that we have two parameter sets that produce closely the same likelihood, which in itself would be relatively straight-forward in a complex LSM if we are looking at multiple outputs due to the cost function determining the likelihood. In that case as we approach these parameter sets, the likelihood function will continue to be univariate and smooth in a similar fashion even if driven by two different sets.

This isn't to argue against using likelihood for the emulator, rather that I don't understand how it solves any of the issues raised before it?

This sentence cites the study by Fer et al. (2018) to demonstrate that emulators need not describe the whole model but simply what we are interested in. To ease confusion, we have removed the end of the sentence.

Line 116: "2.2.2 Variational data assimilation"

In addition for the VarDA application here seemingly being 4D Var without ever directly identified as that, there is also no discussion at all in the methodology section about the required adjoint version of the model. Which in turn is a bit confusing as at least the gradient based algorithm used for the calibration is based on the assumption that all the information is transferred to the same point of time in order calculate the gradient.

Now I do realize there have been ways to sidestep this in previous publications discussing why the adjoint was not used and even here it is mentioned briefly in the discussion. However, even if choosing such approaches, the methods section should be transparent about that as well as about the reasoning behind the choice. This is especially important here because the previous works that have avoided the adjoint have worked with the simpler system than what is done here. Which naturally raises the question that if some of the challenges the VarDA is having are at least partially due to skipping that part of the process as the required assumptions no longer hold as strongly.

We apologise for being more transparent in this section. Deriving the tangent linear or adjoint of complex models like ORCHIDEE is a huge challenge, given the over-evolving nature of these models. A tangent linear does exist for outdated version of ORCHIDEE, and so instead we rely on finite-differences. There is an ongoing effort at LSCE to derive the tangent linear for any version of ORCHIDEE, but this has been a huge undertaking and we do not currently have tangent linear for the current version of ORCHIDEE.

The reviewer is correct that what we do is a form of 4D-Var. As such, referring to the technique as 4D-Var has added to L32:

Furthermore, we often rely on **a 4-dimensional** variational DA (**simply referred to here as** VarDA) **framework**, in which all observations within the assimilation time-window are used to create a cost function which is then minimised.

And to L116:

**Here we treat z, the observations, as a vector to assimilate over the whole time window. This is known as 4DVar (compared to 3DVar where the observations are compared to a single model output at a time).**

Furthermore, we have added the following text to L128:

**Note that here we use the term variational to describe the form of the cost function minimised. While the classical approach to minimise this function relies on gradient-based methods, in the absence of gradient information, other methods have increasingly been used to find the optimum. This has led perhaps to an abusive use of term "variational", however, we feel here it helps to group the two minimisation approaches we wish to compare to the history matching approach.**

L132:

.. referred to as BFGS. **To calculate the gradient information needed for this method, we use finite differences (i.e., the ratio of change in model output against the change in model parameter). While the gradient can be more accurately computed with the tangent linear (linear derivative of the forward model) or adjoint (a computationally efficient way used to calculate the gradient of the cost function), these are extremely hard to compute for complex models like ORCHIDEE and therefore not available at this time.**

Line 174: "The value of *a* is often…"

This part was a bit confusing to me as is *a* set as three here or not? If it is, just state so instead of writing how it is generally done.

Changed to state *a* is set as 3 in this work.

Line 201: "…standard deviation set to 0.1 times the time series' mean."

Why the time series mean? Especially if you are looking at seasonal variables where the assumption would be that the uncertainty is at least partially relative to the measured valued?

In the set-up we use, the error standard deviation is treated as a constant over the whole year (now clarified in the text in accordance with Reviewer 2). As such, using the mean seemed a sensible choice to help generate the Gaussian noise.

Line 203: "…where prior uncertainty is set to 100 % of the parameter range of variation in order to allow for maximal space exploration."

I don't quite understand this as the prior uncertainty is still a normal distribution, correct? So if in this case it is set to 100 %, that would imply the variation in the matrix, which in turn would only cover approximately two thirds of the parameter range with the rest of the uncertainty being beyond that? Reading through the sentence multiple times I think I get what it is stating, in that you are exploring the whole range of allowed parameters, but that is how calibration should work anyway based on prior knowledge, shouldn't it?

Furthermore, if I am understanding this correct, you are essentially setting the prior uncertainty so high that the prior state doesn't really affect the calibration? Which again raises the question why even do this experiment with VarDA as you are here handicapping against its strengths?

The reviewer is correct - the prior uncertainty is still a normal distribution. Nevertheless, the background term still impacts the calibration - this high range of variations means its weight is weakened (compared to past studies) with this formulation but not ignored. This comment was more with regards to past experiments where a fraction of the range is used (e.g., 40% in Kuppel et al., 2012). We have added this to the text:

> where prior uncertainty is set to 100 % of the parameter range of variation **(compared to 40% of the range used in Kuppel et al., 2012)** in order to allow for maximal space exploration.

By having this increased range, we do put less weight on the background making the stochastics experiments more comparable (i.e., the experiments starting at 200 different locations). Finally, even though the prior uncertainty is a normal distribution, allowing for 100% the prior range does let us test values right near the edges (in the tail of the normal distributions) making the GA and BFGS experiment more comparable with the History Matching experiment where again the full parameter space is considered.

Line 206: "Here, we focus on the first year…"

You are using data from only one year to calibrate parameters affecting seasonally changing parameter values? Why not use multiple years? Especially since you are running a twin experiment which involves creating a synthetic time series anyway?

It is true that we could have focused on more than one year in the calibration, but we felt that one year was sufficient to explore the different methods and was easier to illustrate in the figures. Furthermore, increasing the number of years would have increased the computational cost of the study, which would have rendered the "stochastic" experiments (i.e., when we perform 200 optimisations for each minimisation algorithm) very heavy. We have added the following to the text:

> Here, we focus on the first year of the time series (year 2005) for calibration **(to save on computational cost)** and the rest of the time series (years 2006-2009) for evaluation.

Line 241: "3. Results"

I will write a generic comment on this section instead of raising the issue line by line. One can have a "Results and discussions" section or one can have them separately. Here, though, despite there being those individual sections, there are multiple points in the results were amidst presenting the experiment, there are lines which theorize on the meaning of the results. Essentially text that belongs to discussion where it could be contextualized together instead of the scattered approach here.

Thus my suggestion is to going through this whole section and consider moving the lines about the implications of the results to the discussion section. Not only will it make the results section easier to read, it will also allow a more concrete analysis of what can deducted from the experiments.

Although we do see the merit in having a clearer separation between the Results and the Discussion, as RC2 comments, this is a matter of style. We believe that discussing the key interpretation with the results and then having a broader discussion of the pro and cons of the methods in the "Discussion" section works well for this type of comparison study.

Line 322: "This is further reduced to less…"

This is a bit unclear as the sentence before states that the cutoff value was reduced by over 80 % during the first pass with the following part being about how it went to less than 10 % after ten iterations. So is the 10 % referenced here from the cutoff value? Also wouldn't this imply that the benefit of the further iterations was not that great as the first attempt already reduced uncertainty to near the final result?

I realize later on it is stated that after the 5th wave the improvements were marginal, but it is not obvious to me what is used as the standard for that here?

There may be some confusion here between the cutoff, a number we control but never actually change in this work, which is how we decide parameters are too far from our target, and the Xnroy, which is the not-ruled-out-yet space, which we are trying to reduce with the different experiments. We never change in the cutoff in this work since a value of 3 can be justified through the 3sigma rule (L174), and reducing its value needs more justification. In these two sentences, we discuss how the Xnroy has been successfully reduced, first by 80% and then by over 90%.  We have added the following to hopefully alleviate some of the confusion:

> This is further reduced to less than 10% **of the original parameter space** remaining by the end of the tenth wave.

It is true that in this example, most of the Xnroy reduction occurs in the first wave. However, this is not guaranteed to be the case in other examples.

Line 340: "3.2 Implementing process-oriented metrics"

I feel there is a lot of text about implementation and reasoning here that belongs in the methods section.

We have moved the text from this section to the methods.

Now in methods:

> Indeed, using RMSD is often discouraged since it is usually associated to a small signal-to-noise ratio. Furthermore, the implausibility (Eq.~\ref{eq:imp}) is already similar to root-mean-square error \citep{couvreux2021process}.To select informative metrics, it can be helpful to identify specific features we want to constrain. For example, for both the NEE and LE fluxes, we are looking at a seasonal cycle. As such, we expect NEE to have a global sink (i.e., maximum carbon uptake) and LE to have a global peak (i.e., maximum evapotranspiration) in summer. As well as constraining the magnitude of these turning points, we might also want to consider constraining when they occur or the rate of change leading to and from them (i.e. the gradient of slopes). Evap\textsubscript{res} and Root\textsubscript{prof} parameters impact the slopes of the LE seasonal curve in spring and autumn, respectively, so focussing on these gradients may help better inform on these parameters. Similarly, L\textsubscript{agecrit} impacts senescence, so the slope in Autumn is of particular interest. We also know that in winter, there will be little to no photosynthesis (since we are considering a deciduous site). Similarly, we expect low rates of terrestrial ecosystem respiration during these months and, therefore, can constrain NEE in winter. This is similar to constraining the initial carbon pools in the model. In this work, we consider four of these metrics: i) Min/Max of the seasonal cycle (sink for NEE, peak for LE), ii) the slope during spring (taken as the difference between April and February monthly means) and iii) the slope during the senescence period (taken as the difference between September and August monthly means), and iv) initial carbon stocks (NEE only).

Left in Sect. 3.2:

> One of the strengths of HM is that we can easily apply different metrics and so in this section, we consider the additional constraints these metrics bring.

Line 404: "…we compare it to VarDA typically…"

Again just disagreeing with the argument that VarDA is the typical approach used for model calibration even with LSM.

Hopefully the arguments above will help the reviewer understand our position. Nevertheless, we have soften the statement as follows:

> "we compare it to VarDA  **often** used for parameters estimation in land surface models **(especially when calibrating the ORCHIDEE land surface model)**."

Line 443: "The example we test here is illustrative, but very simple."

First, again to be nitpicky, but I think it should be simple, but illustrative.

Changed

Second, and more to the actual commentary, I am not quite certain I understand how this is illustrative as the manuscript tests the methodology with data from a single site. So there is not that much yet that can be said about the multisite performance, especially because those have their own challenges. While it is true that they usually address, at least partially, the equifinality issue, they also do lead to wider parameter uncertainty distributions as there are dynamics at various sites that are not explicitly included in the models. Which in turn is a wider question about how would HM perform in those circumstances that the results here do not give insight into.

Apologies for the confusion. It is true that the way the paragraph is currently written, one infers that through the simple and illustrative test of the paper, we can gain insights into a multisite experiment. Instead, we were trying to suggest that a next step would be a multisite experiment, which in itself has advantages, especially dealing with some of the equifinality issues highlighted in this work. The reviewer is correct that multisite optimisation also have drawbacks - to address these we usually careful select sites for these types of optimisation to ensure they have similar behaviours and that we do not keep any outliers impacting the results. We have rewritten the paragraph as follows to distintangle the separate ideas:

> The example we test here is  very simple**, but illustrative**. **Nevertheless, we only test one site and as we have seen, this suffers from a high degree of equifinality.** Several studies have shown moving from a single site setup to multisite optimisations (i.e., finding one common set of parameter for multiple sites) results in more robust optimisation, less likely to get stuck in local minima (Kuppel et al., 2012; Raoult et al., 2016; Bastrikov et al., 2018). This is because the multiple constraints from the different sites smooth out parameter space. **Even so, multisite optimisation can suffer from wider parameter uncertainty distributions as there are dynamics at various sites that are not explicitly included in the models, necessitating a robust preselection of the subset of sites used in the experiment. To further test the strength of the HM method, a new step will be to move** to **a** multisite **setup**. **We anticipate that one of** the  strength of HM,  its use of emulators, will become more apparent when we move to **these more cost experiments. Furthermore, emulators will greatly benefit the optimisation of**  larger regions and more costly processes (e.g., spin up). Finally, although the twin experiment is very informative, the next **key** step will be to use real world data to assess the full potential of HM.

Line 470: "The true strength of HM is its ability to identify structural issues."

This is a tricky one. I don't agree with this, however, at the same time I feel this might be more of a terminological question. For me, structural issues relate more to the actual model equations and included dynamics, which HM in itself is not any better in isolating than any other calibration method. Don't get me wrong, HM is a useful testing tool in such situation, but I would not that more than anything else as it is still relying on the expertise of the user.

If, though, the structural error here is more in reference in how we set the priors of the parameter values themselves, then there I would agree that that is something where application of HM methods have value.

The review makes fair point, and therefore we have chosen the soften the statement as follows:

> "The true strength of HM is its ability to ability **to help with the identification** of structural issues **(Couvreux et al., 2021).**"

Just to expand a bit on the motivation behind the statement: if everything is ruled out (as long as the emulator is accurate), it tells you that your model is not able to match the observations up to current tolerances to error, and as Var[e] is usually somewhat known, the issue is you didn't properly account for the discrepancy, i.e. structural error. Therefore, it can tell you something about the presence of structural issues - although these can be hard to diagnose if RMSD is the metric, and it does require some expertise to interpret how it is going wrong. When compared to optimisation (single best x) or probabilistic calibration (distribution over x) - both these approaches are guaranteed to return a 'best' x or a distribution over x, regardless of whether you've specified your errors correctly. If you examine these results, you could then determine that actually you're outside the error tolerances, and need to change them, as with HM. One can argue that you can therefore get to the same result, just with an extra step, if you decide to do it. It basically boils down to a philosophical difference - clearly saying all model runs are outside tolerance, vs post-hoc deciding that your 'best' or your distribution is not good enough.

**References**

Bacour, C., MacBean, N., Chevallier, F., Léonard, S., Koffi, E. N., & Peylin, P. (2023). Assimilation of multiple datasets results in large differences in regional- to global-scale NEE and GPP budgets simulated by a terrestrial biosphere model. *Biogeosciences, 20*(1089–1111). https://doi.org/10.5194/bg-20-1089-2023

Bannister, R. N. (2017). A review of operational methods of variational and ensemble-variational data assimilation. *Quarterly Journal of the Royal Meteorological Society, 143*(607–633). https://doi.org/10.1002/qj.2982

Bastrikov, V., MacBean, N., Bacour, C., Santaren, D., Kuppel, S., & Peylin, P. (2018). Land surface model parameter optimisation using in situ flux data: Comparison of gradient-based versus random search algorithms (a case study using ORCHIDEE v1.9.5.2). *Geoscientific Model Development, 11*(4739–4754). https://doi.org/10.5194/gmd-11-4739-2018

Castro-Morales, K., Schürmann, G., Köstler, C., Rödenbeck, C., Heimann, M., & Zaehle, S. (2019). Three decades of simulated global terrestrial carbon fluxes from a data assimilation system confronted with different periods of observations. *Biogeosciences, 16*(3009–3032). https://doi.org/10.5194/bg-16-3009-2019

Couvreux, F., Hourdin, F., Williamson, D., Roehrig, R., Volodina, V., Villefranque, N., Rio, C., Audouin, O., Salter, J., Bazile, E., & Brient, F. (2021). Process-based climate model development harnessing machine learning: I. A calibration tool for parameterization improvement. *Journal of Advances in Modeling Earth Systems, 13*(3), e2020MS002217. https://doi.org/10.1029/2020MS002217

Fairbairn, D., Pring, S. R., Lorenc, A. C., & Roulstone, I. (2014). A comparison of 4DVar with ensemble data assimilation methods. *Quarterly Journal of the Royal Meteorological Society, 140*(281–294). https://doi.org/10.1002/qj.2135

Fer, I., Kelly, R., Moorcroft, P. R., Richardson, A. D., Cowdery, E. M., & Dietze, M. C. (2018). Linking big models to big data: Efficient ecosystem model calibration through Bayesian model emulation. *Biogeosciences, 15*(5801–5830). https://doi.org/10.5194/bg-15-5801-2018

Fisher, R. A., & Koven, C. D. (2020). Perspectives on the future of land surface models and the challenges of representing complex terrestrial systems. *Journal of Advances in Modeling Earth Systems, 12*, e2018MS001453. https://doi.org/10.1029/2018MS001453

Han, Y., Lu, C., Sun, Q., & Zhang, H. (2024). Surrogate model for geological CO2 storage and its use in hierarchical MCMC history matching. *Advances in Water Resources, 187*, 104678. https://doi.org/10.1016/j.advwatres.2024.104678

Hourdin, F., et al. (2017). The art and science of climate model tuning. *Bulletin of the American Meteorological Society, 98*(589–602). https://doi.org/10.1175/BAMS-D-15-00135.1

Hourdin, F., Ferster, B., Deshayes, J., Mignot, J., Musat, I., & Williamson, D. (2023). Toward machine-assisted tuning avoiding the underestimation of uncertainty in climate change projections. *Science Advances, 9*(29), eadf2758. https://doi.org/10.1126/sciadv.adf2758

Kaminski, T., Knorr, W., Scholze, M., Gobron, N., Pinty, B., Giering, R., & Mathieu, P. P. (2012). Consistent assimilation of MERIS FAPAR and atmospheric CO2 into a terrestrial vegetation model and interactive mission benefit analysis. *Biogeosciences, 9*(3173–3184). https://doi.org/10.5194/bg-9-3173-2012

Kaminski, T., Knorr, W., Schürmann, G., Scholze, M., Rayner, P. J., Zaehle, S., Blessing, S., Dorigo, W., Gayler, V., Giering, R., Gobron, N., Grant, J. P., Heimann, M., Hooker-Stroud, A., Houweling, S., Kato, T., Kattge, J., Kelley, D., Kemp, S., Koffi, E. N., Köstler, C., Mathieu, P. P., Pinty, B., Reick, C. H., Rödenbeck, C., Schnur, R., Scipal, K., Sebald, C., Stacke, T., Van Scheltinga, A. T., Vossbeck, M., Widmann, H., & Ziehn, T. (2013). The BETHY/JSBACH Carbon Cycle Data Assimilation System: Experiences and challenges. *Journal of Geophysical Research: Biogeosciences, 118*(1414–1426). https://doi.org/10.1002/jgrg.20118

King, R. C., Mansfield, L. A., & Sheshadri, A. (2024). Bayesian history matching applied to the calibration of a gravity wave parameterization. *Journal of Advances in Modeling Earth Systems, 16*, e2023MS004163. https://doi.org/10.1029/2023MS004163

Knorr, W., Kaminski, T., Scholze, M., Gobron, N., Pinty, B., Giering, R., & Mathieu, P.-P. (2010). Carbon cycle data assimilation with a generic phenology model. *Journal of Geophysical Research: Biogeosciences, 115*, G04017. https://doi.org/10.1029/2009JG001119

Kuppel, S., Peylin, P., Chevallier, F., Bacour, C., Maignan, F., & Richardson, A. D. (2012). Constraining a global ecosystem model with multi-site eddy-covariance data. *Biogeosciences, 9*(10), 3757–3776. https://doi.org/10.5194/bg-9-3757-2012

Kuppel, S., Peylin, P., Maignan, F., Chevallier, F., Kiely, G., Montagnani, L., & Cescatti, A. (2014). Model–data fusion across ecosystems: From multisite optimizations to global simulations. *Geoscientific Model Development, 7*(6), 2581–2597. https://doi.org/10.5194/gmd-7-2581-2014

Maschio, C., & Schiozer, D. J. (2014). Bayesian history matching using artificial neural network and Markov Chain Monte Carlo. *Journal of Petroleum Science and Engineering, 123*, 62–71. https://doi.org/10.1016/j.petrol.2014.08.014

Mohamed, L., Azzali, S., Angelini, S., & Bertino, L. (2012). Population MCMC methods for history matching and uncertainty quantification. *Computational Geosciences, 16*, 423–436. https://doi.org/10.1007/s10596-012-9297-7

Peylin, P., Bacour, C., MacBean, N., Leonard, S., Rayner, P., Kuppel, S., Koffi, E., Kane, A., Maignan, F., Chevallier, F., & Ciais, P. (2016). A new stepwise carbon cycle data assimilation system using multiple data streams to constrain the simulated land surface carbon cycle. *Geoscientific Model Development, 9*(9), 3321–3346. https://doi.org/10.5194/gmd-9-3321-2016

Pinnington, E., Quaife, T., Lawless, A., Williams, K., Arkebauer, T., & Scoby, D. (2020). The Land Variational Ensemble Data Assimilation Framework. *Geoscientific Model Development, 13*, 55–69. https://doi.org/10.5194/gmd-13-55-2020

Raoult, N. M., Jupp, T. E., Cox, P. M., & Luke, C. M. (2016). Land-surface parameter optimisation using data assimilation techniques: The adJULES system V1.0. *Geoscientific Model Development, 9*(8), 2833–2852. https://doi.org/10.5194/gmd-9-2833-2016

Rayner, P. J., Michalak, A. M., & Chevallier, F. (2019). Fundamentals of data assimilation applied to biogeochemistry. *Atmospheric Chemistry and Physics, 19*(13911–13932). https://doi.org/10.5194/acp-19-13911-2019

Rayner, P. J., Scholze, M., Knorr, W., Kaminski, T., Giering, R., & Widmann, H. (2005). Two decades of terrestrial carbon fluxes from a carbon cycle data assimilation system (CCDAS). *Global Biogeochemical Cycles, 19*(2). https://doi.org/10.1029/2004GB002254

Santaren, D., Peylin, P., Viovy, N., & Ciais, P. (2007). Optimizing a process-based ecosystem model with eddy-covariance flux measurements: A pine forest in southern France. *Global Biogeochemical Cycles, 21*. https://doi.org/10.1029/2006GB002834

Schürmann, G. J., Kaminski, T., Köstler, C., Carvalhais, N., Voßbeck, M., Kattge, J., Giering, R., Rödenbeck, C., Heimann, M., & Zaehle, S. (2016). Constraining a land-surface model with multiple observations by application of the MPI-Carbon Cycle Data Assimilation System V1.0. *Geoscientific Model Development, 9*, 2999–3026. https://doi.org/10.5194/gmd-9-2999-2016

Scholze, M., Kaminski, T., Knorr, W., Blessing, S., Voßbeck, M., Grant, J. P., & Scipal, K. (2016). Simultaneous assimilation of SMOS soil moisture and atmospheric CO2 in-situ observations to constrain the global terrestrial carbon cycle. *Remote Sensing of Environment, 180*, 334–345. https://doi.org/10.1016/j.rse.2016.02.060

Scholze, M., Kaminski, T., Rayner, P., Knorr, W., & Giering, R. (2007). Propagating uncertainty through prognostic carbon cycle data assimilation system simulations. *Journal of Geophysical Research: Atmospheres, 112*(D17). https://doi.org/10.1029/2007JD008642

Scholze, M., Kaminski, T., Knorr, W., Voßbeck, M., Wu, M., Ferrazzoli, P., Kerr, Y., Mialon, A., Richaume, P., Rodríguez-Fernández, N., & Vittucci, C. (2019). Mean European carbon sink over 2010–2015 estimated by simultaneous assimilation of atmospheric CO2, soil moisture, and vegetation optical depth. *Geophysical Research Letters, 46*(23), 13796–13803. https://doi.org/10.1029/2019GL085725

Verbeeck, H., Peylin, P., Bacour, C., Bonal, D., Steppe, K., & Ciais, P. (2011). Seasonal patterns of CO2 fluxes in Amazon forests: Fusion of eddy covariance data and the ORCHIDEE model. *Journal of Geophysical Research: Biogeosciences, 116*, G02018. https://doi.org/10.1029/2010JG001544

Williamson, D., Blaker, A. T., Hampton, C., & Salter, J. (2015). Identifying and removing structural biases in climate models with history matching. *Climate Dynamics, 45*, 1299–1324. https://doi.org/10.1007/s00382-014-2378-z

---

## Author Response (AR2)

We are very grateful to both reviewers for taking the time to read the revised manuscript once more and for the recommendations for publication.

**Reviewer 1**

This is the review for the revised manuscript "Exploring the Potential of History Matching for Land Surface Model Calibration".

Overall I am quite satisfied with the adjustments and changes made in response to my review of the previous version. To be honest, I still have fundamental reservations about how informative the comparisons between the VarDA and HM methods are, but it touches perhaps more on certain philosophical aspects of this particular field. And I can not deny that there isn't a history of VarDA use in these calibrations nor that even with that particular issue, the manuscript does excellent work with HM application. So it would be unfair of me to make such an issue a stumbling stone here as I can see another reviewer being fine with it.

Due to this, I am recommending the manuscript to be accepted as is.

Thank you for the recommendation. It is true that since this study intersects several disciplines, there are still philosophical debates around terminology and best practice methods. Hopefully these issues will be more fully addressed in future works.

**Reviewer 2**

Two very minor comments:
Thank yu for the comments, these have been addressed.

Line 31 - I suggest a slight rearrangement and addition to the sentence: "In contrast, in climate studies where we tend to be more interested in long-term trends, we rely less on initial state optimisation and more on parameter calibration or state adjustment. This is espectially true for carbon cycle models, where, in addition, a lot of processes are based on empirical equations that may not be perfect representations of the actual processes.
Done

Line 355 - remove word 'remaining' to match response to reviewers
Done